# Single-cell mRNA-regulation analysis reveals cell type-specific mechanisms of type 2 diabetes

Perturbed secretion of insulin and other pancreatic islet hormones is the main cause of type 2 diabetes (T2D). The islets harbor five cell types that are potentially altered differently by T2D. Whole-islet transcriptomics and single-cell RNA-sequencing (scRNAseq) studies have revealed differentially expressed genes without reaching consensus. Here, we demonstrate that further insights into T2D disease mechanisms can be obtained by network-based analysis of scRNAseq data from individual cell types. We developed differential gene coordination network analysis (dGCNA) and analyzed islet SmartSeq2 scRNAseq data from 16 T2D and 16 non-T2D individuals. dGCNA reveals T2D-induced cell type-specific networks of dysregulated genes with remarkable ontological specificity, thus allowing for a comprehensive and unbiased functional classification of genes involved in T2D. In beta cells eleven networks of genes are detected, revealing that mitochondrial electron transport chain, glycolysis, cytoskeleton organization, cell proliferation, unfolded protein response and three networks of beta cell transcription factors are perturbed, whereas exocytosis, lysosomal regulation and insulin translation programs are instead enhanced in T2D. Furthermore, we validated the ability of dGCNA to reveal disease mechanisms and predict the functional context of genes by showing that *TMEM176A/B* regulates beta cell microfilament organization and that *CEPBG* is an important regulator of the unfolded protein response. In addition, when comparing beta- and alpha cells, we found substantial differences, reproduced across independent datasets, confirming cell type-specific alterations in T2D. We conclude that analysis of networks of differentially coordinated genes provides detailed insight into cell type-specific gene function and T2D pathophysiology.

Type 2 diabetes (T2D) is a major health care challenge caused by a combination of insulin resistance in target tissues and insufficient insulin release from islet beta cells. Without doubt, impaired islet function is central in T2D, but the underlying mechanisms are not fully understood[1,2]. Transcriptomic data from whole islet preparations have identified differentially expressed genes in T2D[3–7]. A confounding factor in these studies is that the islets are composed of five functionally distinct endocrine cell types[8]. Single-cell RNA-sequencing (scRNAseq) studies of islet cells have revealed cell type-specific gene expression alterations, but with little overlap between studies[9–15]. Thus, novel computational analysis and functional validation have been identified as crucial for advancing our understanding of T2D islet disease mechanisms[15,16]. Network biology allows for the identification of pathways and regulatory mechanisms by inferring network

✉ e-mail: jens.hjerling-leffler@ki.se; nils.wierup@med.lu.se

modules[17–20]. Disease-related changes can be identified by comparing network structures[21,22]. Differential Network Analysis (DiNA) instead relies on constructing and analyzing the network structure of the dynamics between states[23–26]. Here, we developed a version of DiNA specific for scRNAseq data, called differential Gene Coordination Network Analysis (dGCNA). We show that analyzing networks of differentially coordinated genes is a viable and unbiased approach to infer cell type-specific dynamics of the disease from static data. In beta cells, we identify non-canonical genes and pathways in addition to multiple genes and pathways previously implicated in T2D beta cell dysfunction. Importantly, dGCNA also predicts the functional context of genes. Additionally, we identify genes and pathways involved in alpha cells, confirming that T2D entails cell type-specific gene regulatory changes.

## Results

In this study, we generated and merged two high-depth (-1 M reads/cell) scRNAseq datasets from human pancreatic islets. Dataset 1 was generated from hand-picked islets (with confirmed ability to release insulin) from six non-T2D (HbA1c < 6.0%) and six T2D (HbA1c > 6.5% or diagnosis) brain-dead organ donors. Cells were dissociated and analyzed acutely (within 48 h of collection) using Smart-seq2[27]. After filtering, we analyzed 3645 cells in dataset 1. For dataset 2, we analyzed 4866 cells from ten non-T2D donors and ten patients diagnosed with T2D from commercial sources (Prodo Laboratories Inc.) using the same Smart-seq2 pipeline. Part of dataset 2 (six non-T2D and four T2D

donors) has previously been published[9]. Characteristics of the 32 donors and the number of cells/cell type are presented in Supplementary Data 1. After merging the datasets, Conos integration and clustering revealed eleven distinct cell types expressing known pancreatic cell type markers (Fig. 1A–D). We detected on average 6000 genes in alpha- and beta cells (Fig. S1A), and there was no apparent donor effect on cell type clustering (Fig. S1B). We subsequently analyzed the merged dataset in all analyses.

Gene regulatory networks are typically computed across cells and cell types to identify genes that are co-expressed. In addition, co-expression between genes within a cell (for this, the term "coordination" will be used) is likely an important aspect of gene regulation and cell biology. The identification of coordinated clusters of genes within a cell type would suggest that single-cell transcriptomes represent snapshots of parallel cellular states captured across different phases. We thus hypothesized that altered coordination on a transcriptome scale represents meaningful information about disease biology in T2D. To identify the structure of differentially coordinated gene programs that had either been strengthened (hyper-coordinated) or weakened (de-coordinated) in T2D cells, we developed dGCNA, which harnesses the high-dimensionality and biological variability of single-cell transcriptomics (Fig. S2). In essence, this is a differential network analysis on single cell types, meaning that the differential network is built on the differences in gene-pair coordination between the two states (non-T2D and T2D) (Fig. S2A). We performed statistical comparisons of correlation coefficients between gene pairs in a single cell type at a

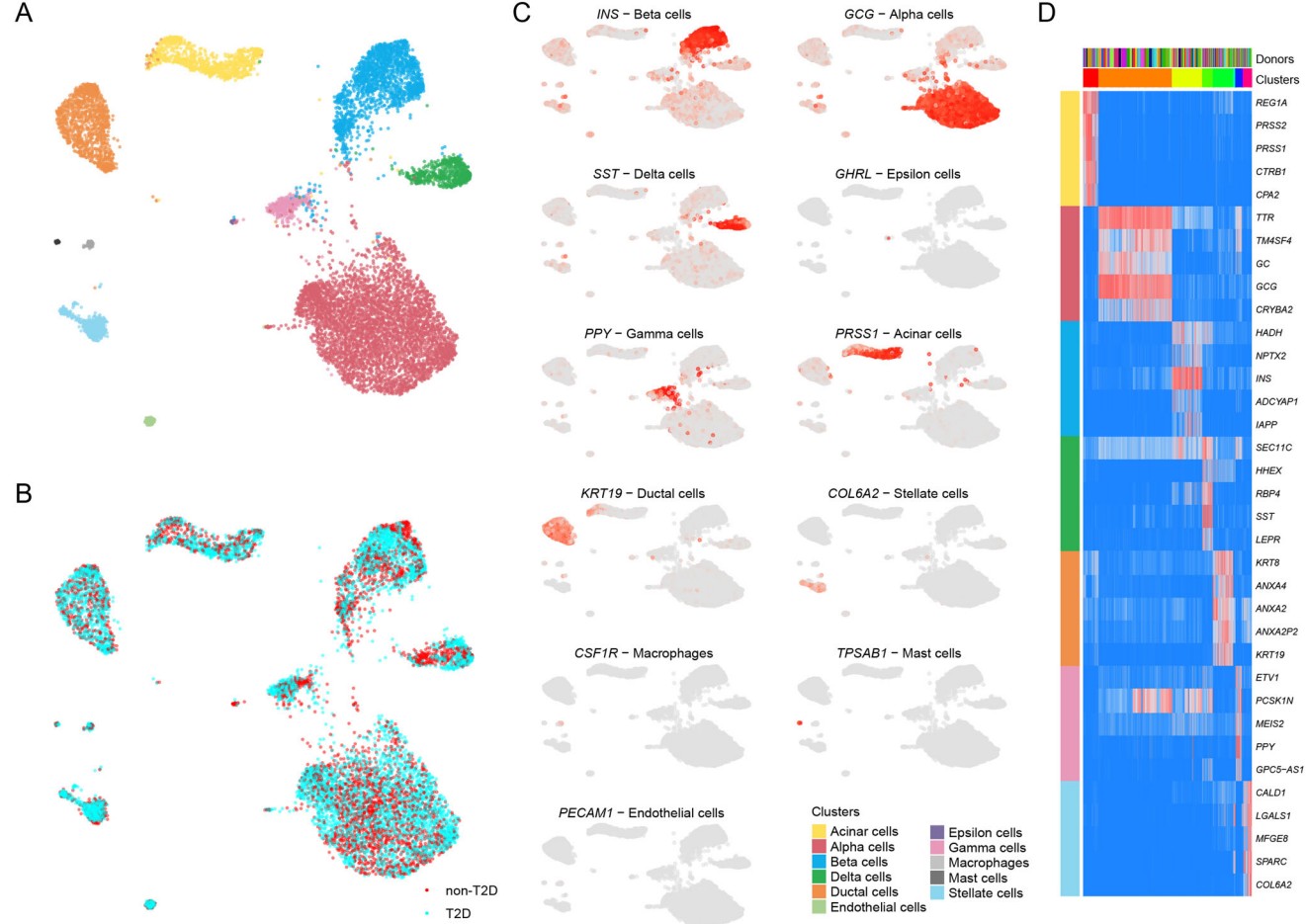

**Fig. 1 | Cell type clustering.** UMAP visualization of 8511 single cells, from 32 donors with cell types (**A**) and diagnosis of donors (**B**) indicated. **C** Expression of known cell type marker genes for all cell types. **D** Top specific (differentially expressed) genes in each cell type. No differentially expressed genes were detected for Epsilon cells, Macrophages, Endothelial cells, or Mast cells, due to the low number of recovered cells. See also Supplementary Data 1 and Fig. S1.

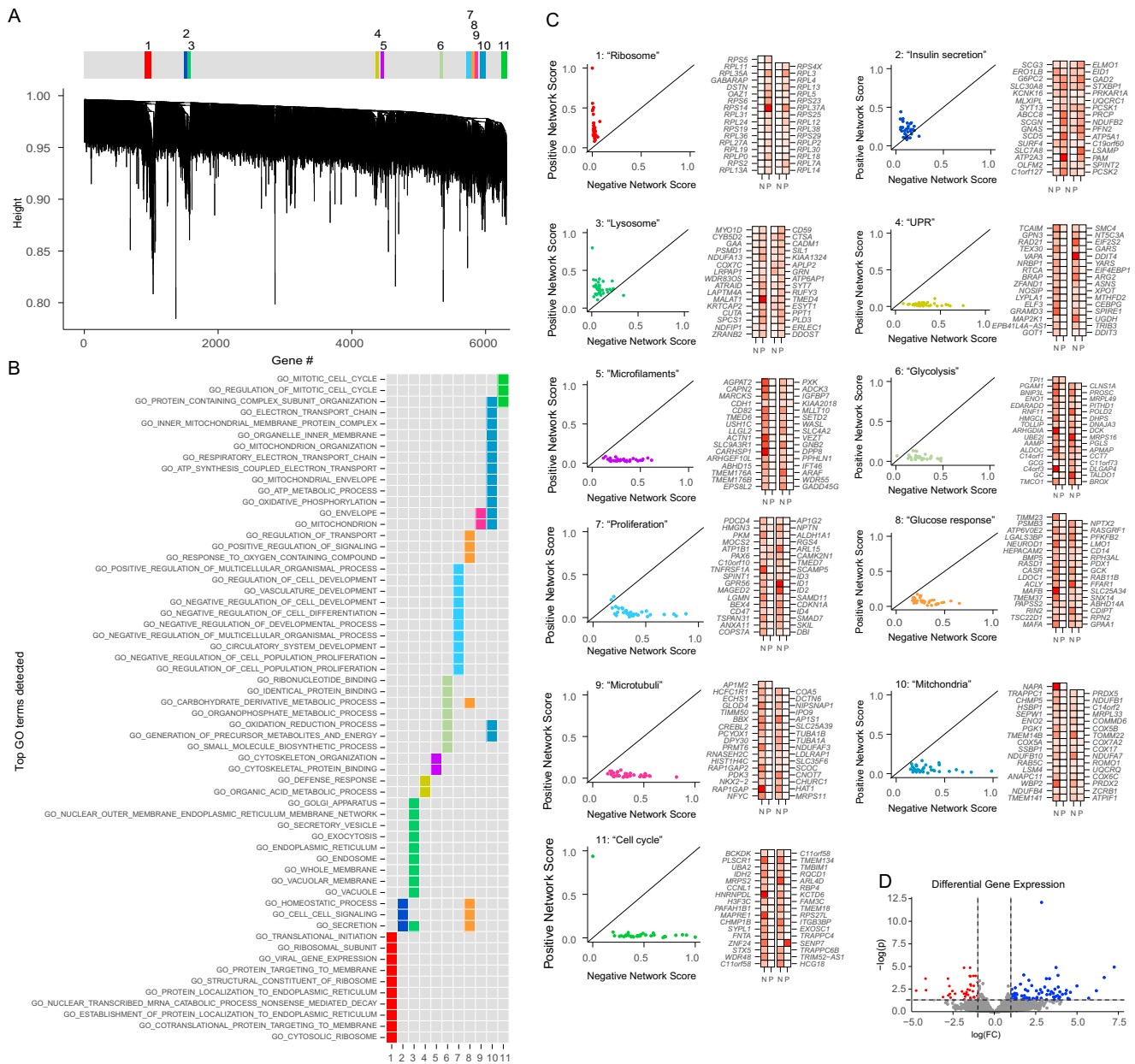

**Fig. 2 | dGCNA uncovers the functional context of genes corresponding to canonical and non-canonical T2D pathways of beta cells. A** Hierarchical clustering dendrogram and clustering of eleven NDCGs in beta cells. Genes with high intramodular connectivity are located at the tip of the module branches since they display the highest interconnectedness with the rest of the genes in the module. Colors represent clusters detected by the dynamic tree cutting algorithm. **B** Plotting the top enriched GO-terms (Biological process) for each NDCG across all NDCGs revealed a high degree of specificity in terms of biological annotation. Columns represent NDCGs and rows are top GO-terms. **C** Scatter plots representing coordination rank scores for genes in each NDCG, with degree of negative (N) and positive (P) scores indicated in red. **D** Plot of differentially expressed genes (pseudo-bulk DESeq2, merged dataset, FDR corrected). See also Supplementary Data 2 and Figs. S2–S4 and S11.

time, using a linear mixed-effect model to account for donor-specific effects (Fig. S2B) and then used a dynamic boot-strap-based threshold to identify robust links to create a robust differential network (RDN) (Fig. S2C). We then performed topological analysis and gene clustering on the RDN (Fig. S2D–F).

To reveal T2D-affected biological pathways in beta cells, we performed dGCNA, and the dendrogram from the topological analysis indicated a high level of modularity. Dynamic tree cutting of the dendrogram revealed eleven networks of differentially coordinated genes (NDCGs) pointing to pathway alterations in T2D beta cells (Fig. 2A and Supplementary Data 2). Each of the NDCGs was associated with highly specific gene ontology (GO) terms (Fig. 2B and

Supplementary Data 2) and was named according to biological functions (either the top GO-terms or function identified after literature review). The following NDCGs (Fig. 2A–C, Supplementary Data 2) were identified: "Ribosome" was composed almost entirely of ribosome subunit genes. "Insulin secretion" contained multiple key genes for insulin secretion, including *G6PC2, ABCC8, SLC30A8, KCNK16, PCSK1,* and *PCSK2*[28–30]. "Lysosome" was adjacent to "Insulin secretion" and contained genes related to lysosomal function, e.g., *GAA, PSAP, CTSA,* and also genes with roles in the regulation of insulin secretion, e.g., *SYT7*[31]*, CD59*[32]*, and KIAA1324*[33]. Unfolded protein response ("UPR") contained stress-related genes including *TRIB3, DDIT3, DDIT4, EIF4EBP1,* and *EIF2S2*. Also, *ARG2*, previously shown to be affected in

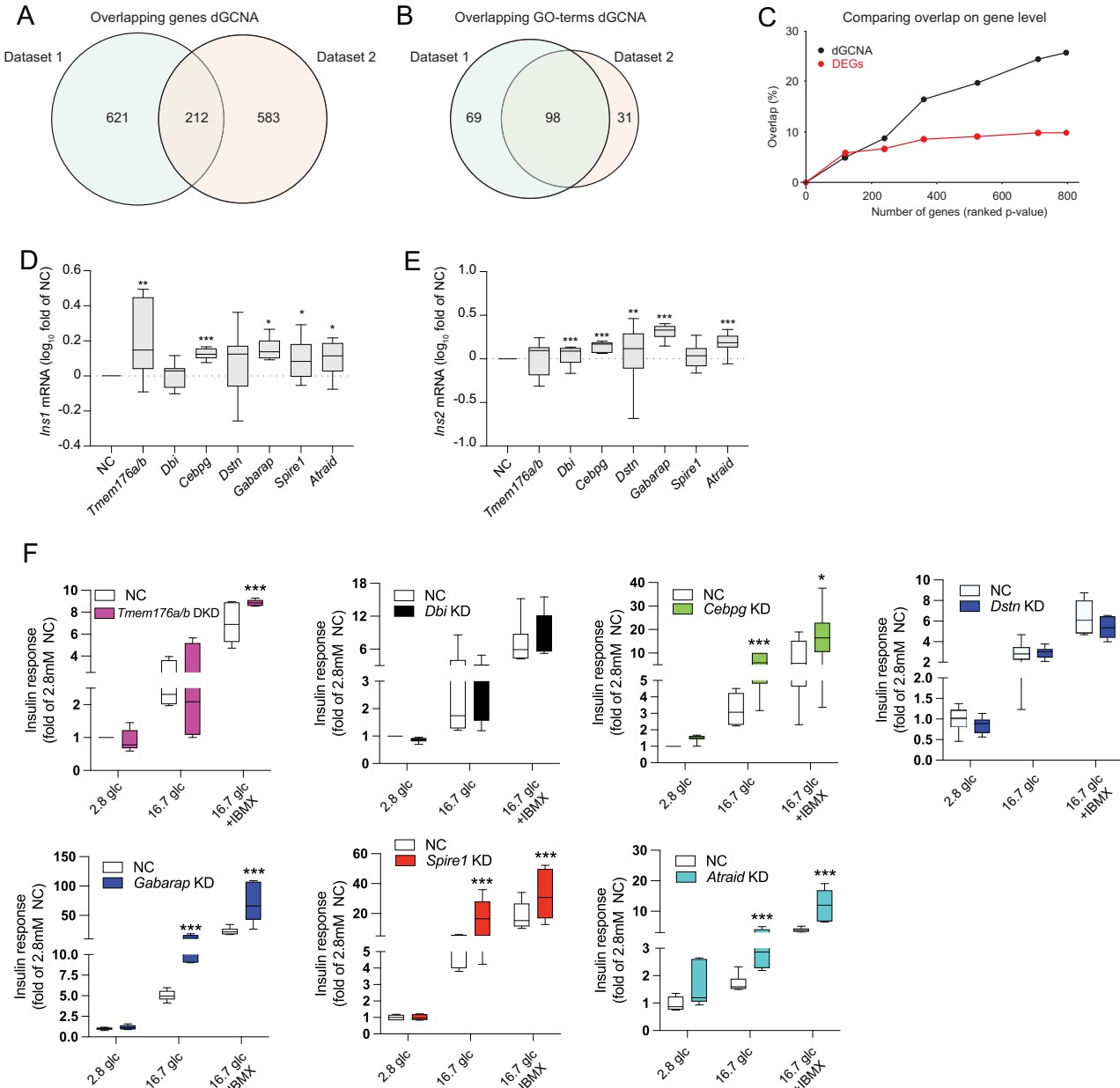

**Fig. 3 | dGCNA outperforms DESeq2 for replication of T2D-affected genes between datasets and the effect of target genes on *Ins1* and *Ins2* mRNA, and on glucose-stimulated insulin secretion. A** Overlap of genes identified with dGCNA in dataset 1 and dataset 2. **B** Overlap of GO-terms for dGCNA genes for dataset 1 and dataset 2. **C** Comparison of overlap of genes between dataset 1 and dataset 2 using DESeq2 and dGCNA on top-ranked 50–800 genes. **D** siRNA knockdown of *Tmem176a/b, Cebpg, Gabarap, Spire1*, or *Atraid* resulted in increased *Ins1* mRNA levels (data presented as $\log_{10}$-fold change of scrambled siRNA control (NC)). **E** Knockdown of *Dbi, Cebpg, Dstn, Gabarap*, or *Atraid* resulted in increased *Ins2* mRNA levels in INS-1 832/13 cells. **F** Knockdown of *Cebpg, Gabarap, Spire1*, or *Atraid*

caused increased insulin secretion in INS-1 832/13 cells at 16.7 mM glucose. At 16.7 mM glucose with IBMX, knockdown of *Tmem176a/b, Cebpg, Gabarap, Spire1*, or *Atraid* resulted in increased insulin secretion. **D–E** $n = 6$ per condition. **F** $n = 6$ per condition except for *Cebpg* ($n = 7$) and *Tmem176a/b* ($n = 5$). Data in (**D–F**) is presented as box plots. Hinges of the box represent the 25th and 75th percentiles, and the line in the box is the median. Whiskers are min and max values. *$p < 0.05$, **$p < 0.01$, ***$p < 0.001$. Two-tailed Student's t-test was used in (**D**) and (**E**). One-way ANOVA, followed by Tukey's post hoc test, was used in (**F**). See also Supplementary Data 4 and Figs. S5–S7. Source data are provided as a Source Data file.

T2D[5], was present here. "Microfilaments" contained genes related to actin organization, e.g., *ACTN1, MARCKS*, and *WASL*. "Glycolysis" contained genes encoding glycolytic enzymes, e.g., *ENO1, ALDOC, PGAM1*, and *TPI1*. "Proliferation" contained *PAX6*[34], *ISL1*[28], and multiple genes involved in cell proliferation and differentiation. "Glucose response" contained several genes crucial for beta cell function, e.g., *PDX1, NEUROD1, MAFA, MAFB, GCK*[35], and *FFAR1*[36]. "Microtubuli" contained genes regulating microtubules and *NKX2.2*[37]. "Mitochondria" was

dominated by genes encoding subunits of complex 1 and 4 of the electron transport chain of the mitochondria. Finally, "Cell cycle" contained mitosis-regulating genes, including *MAPRE1* and *PAFAH1B1*.

To investigate if the NDCGs were either hyper- or de-coordinated we assigned each gene a rank score depending on the eigen-vector centrality (dependent on the number of connections with other genes that were lost or gained) (Fig. 2C). Three NDCGs exhibited a general hyper-coordination in T2D ("Ribosome", "Insulin secretion", and

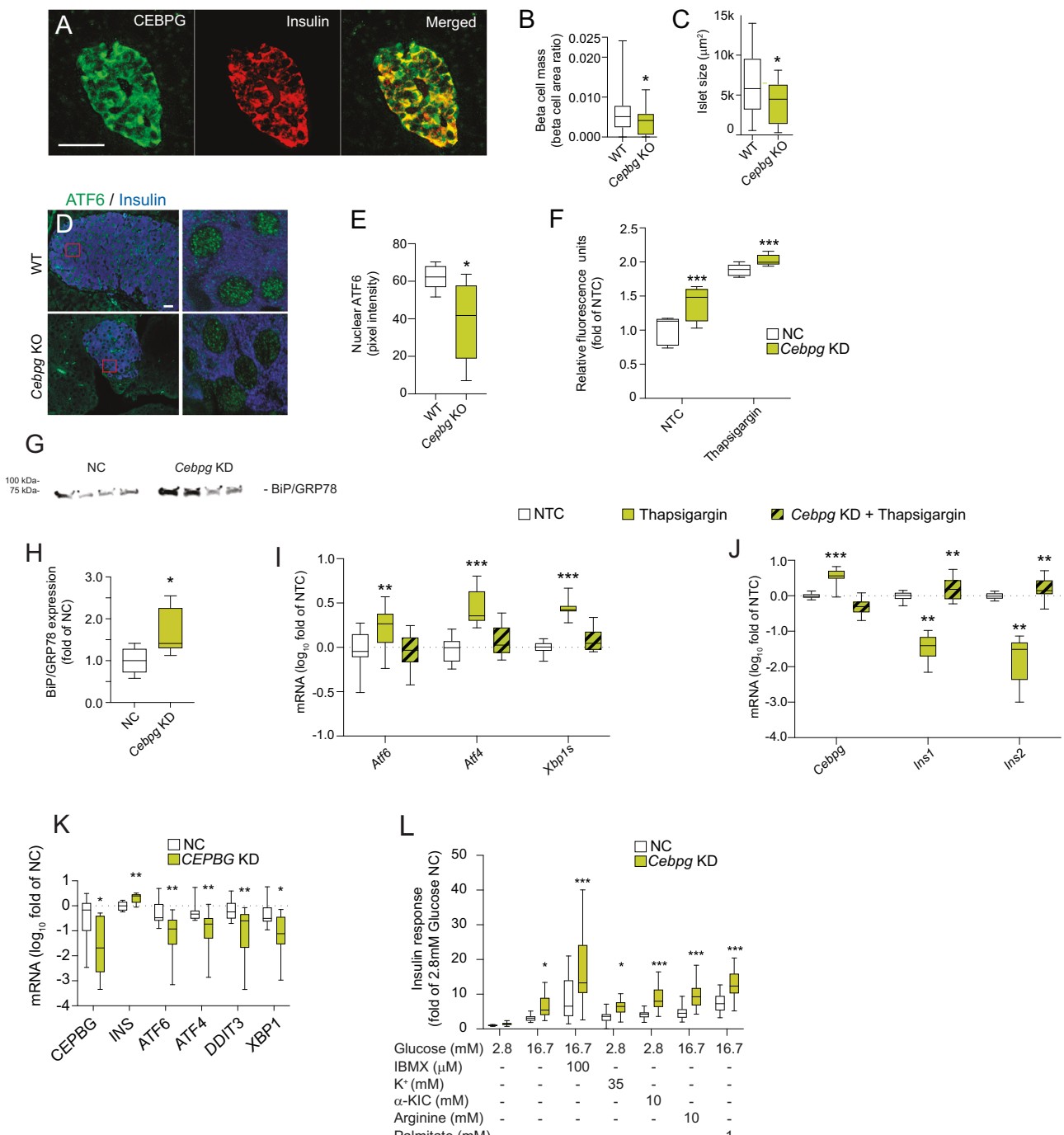

**Fig. 4 | As predicted by dGCNA, *CEBPG* is a regulator of beta cell UPR and a regulator of insulin secretion and gene expression. A** CEBPG protein is expressed in human beta cells. Representative image. 50 islets from 5 donors were analyzed. **B** Reduced beta cell mass and islet size (**C**) in *Cebpg* KO mice (*n* = 6) compared with WT mice (*n* = 16). **D** *Cebgp* KO mice have reduced nuclear expression of ATF6; quantified in (**E**), *n* = 6 per group. **F** Thioflavin assay showing that knockdown of *Cebpg* in INS-1 832/13 cells causes increased protein aggregation, *n* = 5 per condition. **G** Western blot showing that *Cebpg* knockdown in INS-1 832/13 cells increases the expression of BiP/Grp78 (loading control in Fig. S8A); quantified in (**H**), *n* = 6 per condition. **I** Thapsigargin-induced increased expression of *Atf6*, *Atf4*, and *Xbp1s* is abolished by *Cebpg* knockdown in INS-1 832/13 cells, *n* = 6 per condition. **J** Increased expression of *Cebpg* and decreased expression of *Ins1* and *Ins2* in thapsigargin-treated INS-1 832/13 cells. Conversely, *Ins1* and *Ins2* expression

is increased by thapsigargin treatment after *Cebpg* knockdown, *n* = 6 per condition. **K** *CEBPG* knockdown in human islets reduces the expression of the UPR mediators *ATF6*, *ATF4*, *DDIT3*, *and XBP1*, but increases the expression of *INS*, *n* = 5 per condition. **L** *Cebpg* knockdown causes increased insulin secretion from INS-1 832/13 cells stimulated with several different secretagogues, *n* = 7 per condition. NC = scrambled siRNA. NTC = non-treated cells. Data in (**I**, **J**) presented as log fold of NTC. Data in (**B**, **C**, **E**, **F**, and **H**–**L**) is presented as box plots. Hinges of the box represent the 25th and 75th percentiles, and the line in the box is the median. Whiskers are min and max values. *\*p* < 0.05, *\*\*p* < 0.01, *\*\*\*p* < 0.001. Two-tailed Student's *t*-test was used in (**B**, **C**, **E**, and **H**). One-way ANOVA, followed by Tukey's post hoc test, was used in (**F**, **I**–**L**). Scale bars = 50 μm. See also Supplementary Data 5 and Fig. S8-S9. Source data are provided as a Source Data file.

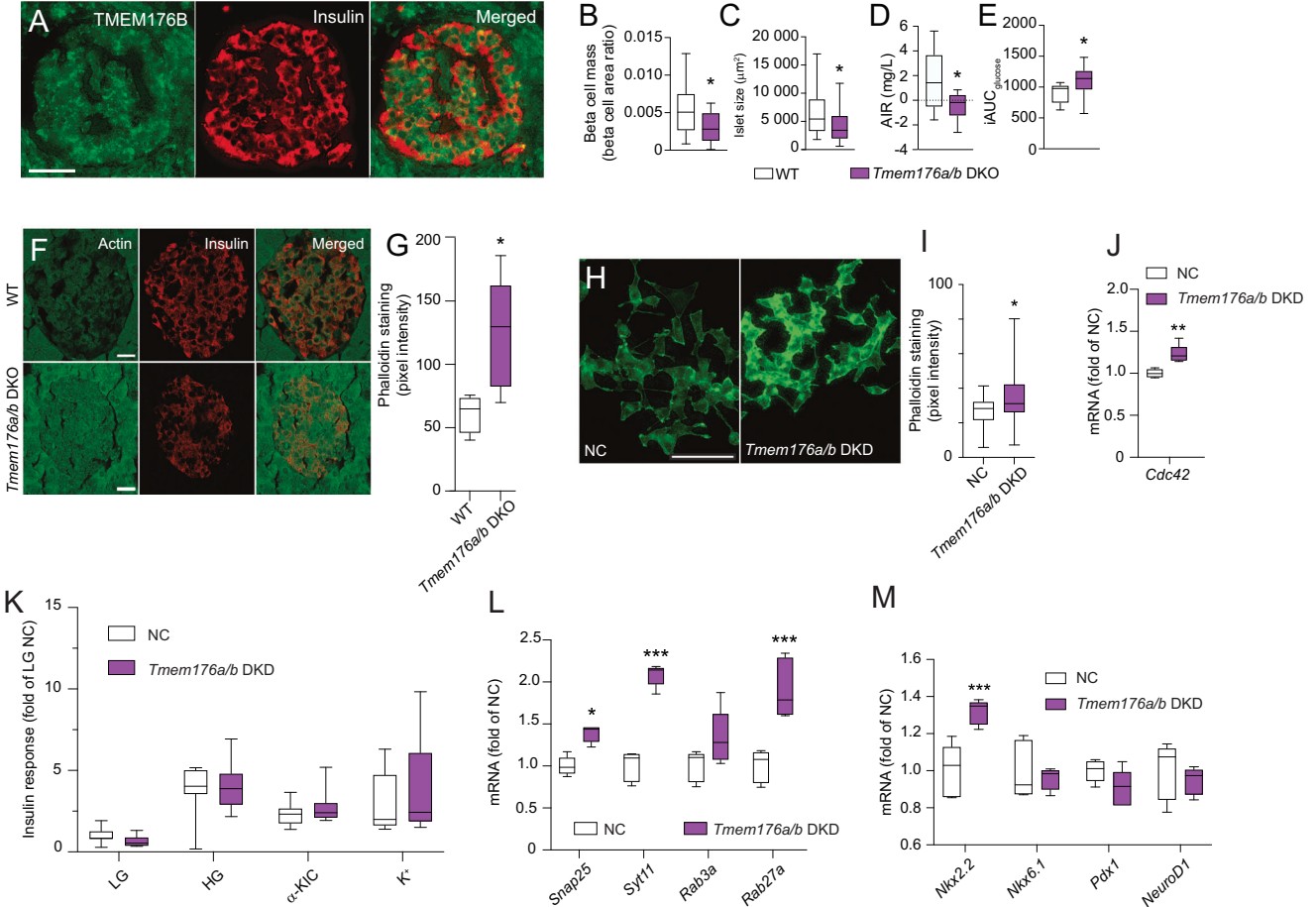

**Fig. 5 | The function of *TMEM176A/B* in beta cells corresponds to the context revealed by dGCNA. A** TMEM176B protein is expressed in human beta cells. Representative image. 50 islets from 5 donors were analyzed. **B** Reduced beta cell mass and islet size (**C**) in *Tmem176a/b* double knockout (DKO) mice (normal diet), *n* = 8 per group. **D** Reduced acute insulin response (AIR) and higher glucose levels calculated as iAUC (**E**) during an IpGTT in *Tmem176a/b* DKO mice fed a high-fat diet (*n* = 10), compared with WT mice fed a high-fat diet (*n* = 10). **F** Phalloidin staining, showing 2-fold increased actin filament density in *Tmem176a/b* DKO mice (*n* = 5) beta cells, compared with WT mice (*n* = 4); quantified in (**G**). **H** Phalloidin staining of INS-1 832/13 cells showing higher actin filament density in *Tmem176a/b* knockdown (DKD) cells; quantified in (**I**), *n* = 6. **J** *Tmem176a/b* knockdown caused increased

mRNA expression of the actin regulator *Cdc42* in INS-1 832/13 cells, *n* = 5. **K** *Tmem176a/b* knockdown had no effect on insulin secretion stimulated with 16.7 mM glucose, 10 mM α-KIC, or 35 mM K⁺ in INS-1 832/13 cells, *n* = 6 per condition. **L** *Tmem176a/b* knockdown caused increased expression of genes encoding key exocytotic proteins and the insulin-regulating transcription factor *Nkx2.2* (**M**) in INS-1 832/13 cells. **L, M** *n* = 5 per condition. Data in (**B–E**, **G**, and **I–M**) is presented as box plots. Hinges of the box represent 25th and 75th percentiles, and the line in the box is the median. Whiskers are min and max values. \**p* < 0.05, \*\**p* < 0.01, \*\*\**p* < 0.001. Two-tailed Student's *t*-test was used in (**B–E**, **G**, and **I–J**). One-way ANOVA, followed by Tukey's post hoc test, was used in (**K–M**). Scale bars = 50 μm. See also Fig. S9-S10. Source data are provided as a Source Data file.

"Lysosome") and eight were overall de-coordinated ("UPR", "Microfilaments", "Glycolysis", "Proliferation", "Glucose response", "Microtubuli", "Mitochondria", and "Cell cycle"). Thus, dGCNA, without prior information, identified specific biological processes, including the most established T2D dysfunction pathways in beta cells (insulin secretion[30], glycolysis[38], mitochondria[39], and UPR[40]), but also genes and processes not previously associated with beta cell dysfunction. Of note, the expression levels of individual genes or clusters of genes did not need to change to be included in NDCGs. Differentially expressed genes are presented in Fig. 2D and Supplementary Data 2.

We calculated enrichment of the modules with genes implicated in beta cell function and T2D via GWAS (Supplementary Data 3). For T2D risk GWAS[41], we saw an enrichment in the module "Insulin Secretion" (p-adj = 4.5e⁻⁶) and remarkably, when limiting the analysis to 14 high-confidence genes (fine-mapped and missense)[42], five of the 32 genes in the module still overlapped with those 14 genes (Odds ratio 129; p-adj = 6.08e⁻⁸) (Fig. S3A, B). For fasting glucose GWAS (https://t2d.hugeamp.org/), we observed enrichment for "Glucose response" (p-adj = 0.018) and "Insulin secretion" (p-adj = 9.2e⁻⁴) (Fig. S3C). For insulin secretion GWAS[43], there was an enrichment in

"Insulin secretion" (p-adj = 9.6e⁻⁴) (Fig. S3D). Next, we compared the genes identified in beta cells using dGCNA with genes identified as regulators of insulin content in a recent whole genome CRISPR screen in EndoC-βH1 cells[44]. 36 (of 671) dGCNA genes, including *NKX2-2*, were found overlapping, but there was no significant enrichment for any module (Supplementary Data 3). Furthermore, 103 of the 671 dGCNA genes were previously shown to correlate with beta cell exocytosis in a Patch-Seq study, and the module "Mitochondria" was significantly (p-adj = 0.002) enriched for genes that negatively correlate with exocytosis in T2D[45] (Fig. S4A and Supplementary Data 3).

To investigate the robustness of dGCNA in the identification of perturbed genes and pathways, we analyzed dataset 1 and dataset 2 separately. We observed 26% overlap in genes within networks with altered coordination in T2D beta cells (Fig. 3A). Furthermore, comparing GO-terms of the genes from networks with altered coordination revealed a striking degree of overlap between the two datasets (Fig. 3B and Fig. S5). Next, we compared dGCNA and differential expression analysis (pseudo-bulk DESeq2, adjusted for age, sex, and BMI) for identifying overlap between the two datasets.

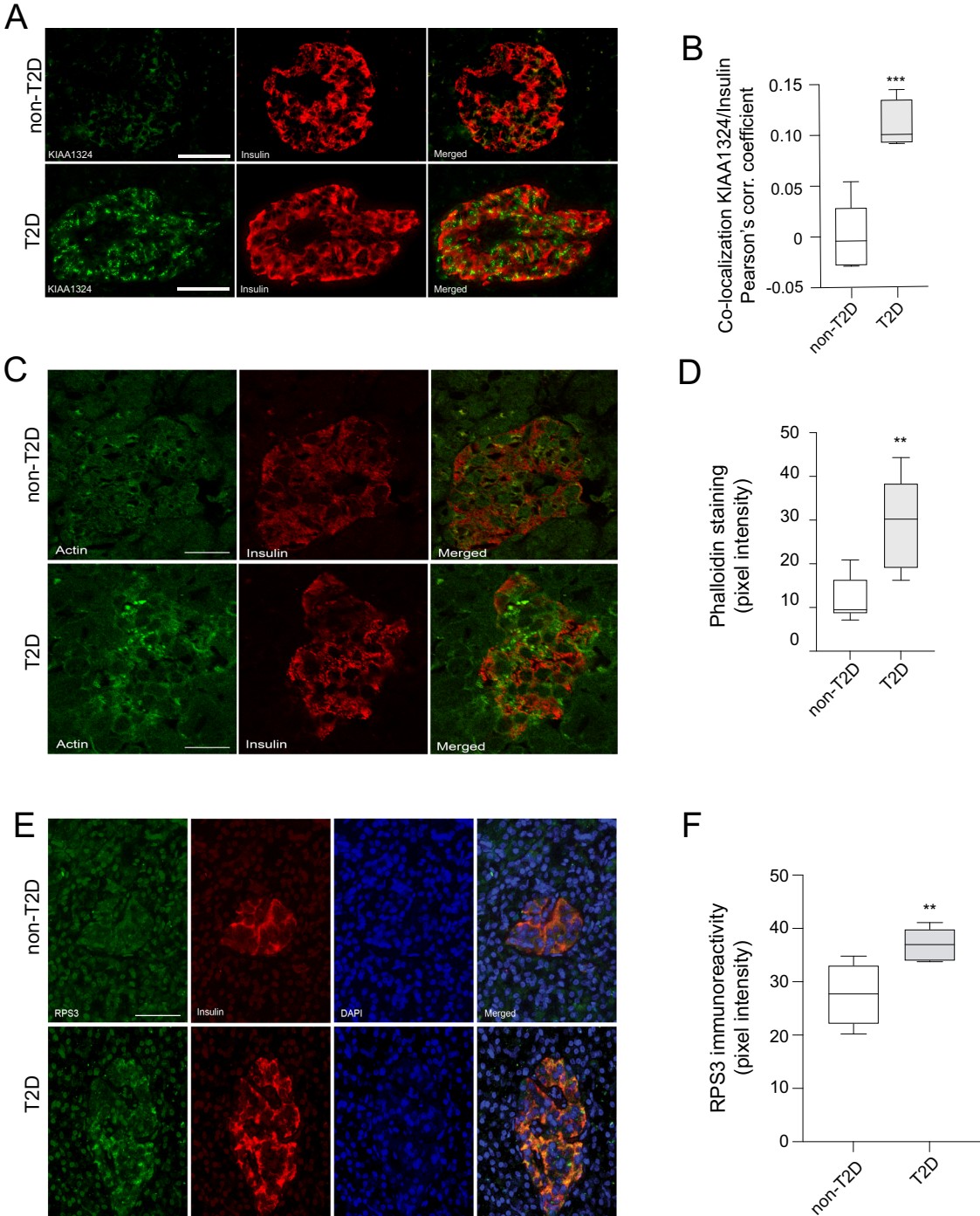

**Fig. 6 | Protein level confirmation of genes and markers of biological processes identified by dGCNA. A** T2D donors have increased beta cell immunoreactivity for KIAA1324 compared with non-T2D donors. **B** Quantification of colocalization of insulin and KIAA1324 in non-T2D ($n = 6$) and T2D donors ($n = 4$). **C** Phalloidin staining illustrating higher beta cell actin filament density in T2D donors compared with non-T2D donors; quantified as pixel intensity in (**D**) in non-T2D ($n = 8$) and T2D donors ($n = 5$). **E** Ribosome staining (RPS3 immunoreactivity) illustrating more ribosome content in T2D donors compared with non-T2D donors; quantified as pixel intensity in (**F**) in non-T2D ($n = 4$) and T2D donors ($n = 6$). Data in (**B**, **D**, and **F**) is presented as box plots. Hinges of the box represent the 25th and 75th percentiles, and the line in the box is the median. Whiskers are min and max values. *$p < 0.05$, **$p < 0.01$, ***$p < 0.001$. Two-tailed Student's $t$-test was used in (**B**, **D**, and **F**). Scale bars = 50 μm. Source data are provided as a Source Data file.

We observed very few significantly differentially expressed genes using DESeq2 (Supplementary Data 2). To compare the methods, we therefore ranked differentially expressed genes based on their $p$-value and compared their overlap up to the number of genes indicated by dGCNA (Fig. 3C). We observed substantially greater overlap using dGCNA as compared to DESeq2. In terms of reproducibility, dGCNA thus outperforms differential expression analysis.

To test the predictive power of dGCNA in identifying genes involved in T2D-related processes, we selected seven genes (*TMEM176A/B* ("Microfilaments"), *DBI* ("Proliferation"), *CEBPG* ("UPR"), *DSTN* ("Ribosome"), *GABARAP* ("Ribosome"), *SPIRE1* ("UPR"), *ATRAID* ("Lysosome")) without a well-known presence or function in human beta cells that were highly ranked in five of the NDCGs (Fig. 2C). For selection criteria, see "Methods" section. We confirmed the presence

of the protein products in human pancreatic sections from donors in dataset 1 (Figs. 4A and 5A and Fig. S6A) and performed morphometric analysis of beta cell protein product expression in T2D *vs.* non-T2D islets, revealing increased protein levels of *DSTN* (Fig. S6B). None of the genes were differentially expressed in our scRNAseq data (Fig. S6C and Supplementary Data 2), and only *SPIRE1* and *ATRAID* were differentially expressed in two out of four previously published bulk RNA-seq datasets[5,7,46] (Supplementary Data 4). For each selected gene, we performed siRNA knockdown (KD) (two siRNAs for each gene) of gene homologs in rat INS-1 832/13 cells (Fig. S7A). KD of each of the seven genes resulted in increased mRNA expression of *Ins1* or *Ins2* (Fig. 3D, E and Fig. S7B). Furthermore, KD of five of the selected genes resulted in affected insulin secretion stimulated with glucose or glucose and IBMX[47] (Fig. 3F). The recently described lysosome-related regulator of beta cell insulin-sensitivity inceptor/*KIAA1324*[33] was highly ranked in "Lysosome," and its protein expression was highly increased in beta cells in pancreatic sections from T2D donors (Fig. 6A, B). Furthermore, in agreement with the two identified NDCGs related to cytoskeletal organization ("Microfilaments" and "Microtubuli"), we confirmed altered microfilament organization, i.e., increased density of phalloidin staining, in beta cells in pancreatic sections from T2D donors (Fig. 6C, D). Moreover, in agreement with the identified ribosomal NDCG, we confirmed altered (increased) ribosomal staining density in T2D beta cells (Fig. 6E, F).

Next, we selected two of the genes for which KO mouse models were available, *CEBPG* (present in the "UPR" NDCG) and *TMEM176A/B* (present in the "Microfilaments" NDCG), that affected insulin secretion and gene expression in INS-1 832/13 cells, for further studies aiming to provide mechanistic understanding of the observed effects, as well as to experimentally validate the predicted function of the genes.

*CEBPG* controls redox homeostasis in normal and cancerous cells[48]. *Cebpg* knockout mice were normoglycemic but had lower beta cell mass and smaller islets compared with WT controls (Fig. 4B, C). As a sign of affected UPR, *Cebpg* knockout mice had less nuclear expression of the UPR regulator ATF6 in beta cells (Fig. 4D, E). Confirming a role for CEBPG in the regulation of UPR, *Cebpg* KD in INS-1 832/13 cells caused increased protein aggregation (Fig. 4F) and GRP78/BiP expression (Fig. 4G, H and Fig. S8A). Furthermore, although *Cebpg* KD had no effect on expression of UPR-mediating genes at basal conditions (Fig. S8B), *Cebpg* KD abolished thapsigargin (inducer of UPR)-induced upregulation of the UPR mediators *Atf4, Atf6,* and *Xbp1s* (Fig. 4I). Thus, suggesting that CEBPG is necessary for UPR and that *Cebpg* KD causes failed UPR and increased ER stress. Furthermore, thapsigargin treatment caused a 4-fold increase in *Cebpg* mRNA expression (Fig. 4J), and *Cebpg* KD reversed the effect of thapsigargin on *Ins1* and *Ins2* mRNA expression in INS-1 832/13 cells (Fig. 4J). A role for CEBPG in the regulation of UPR and downstream insulin gene expression was confirmed in human islets, and *CEBPG* KD caused increased *INS* expression (Fig. 4K) and reduced expression of *ATF4, ATF6,* and *XBP1* (Fig. 4K). Our results thus confirm the dGCNA prediction that CEBPG is a regulator of beta cell UPR, a process indeed known to reduce insulin secretion and production to protect beta cells from death[49,50].

To gain insight into which parts of the insulin secretion machinery that is affected by *Cebpg* KD, we assessed the effect of insulin secretion stimulated with an array of insulin secretagogues (α-ketoisocaproic acid, an activator of mitochondrial metabolism; K⁺, inducing depolarization of the cell membrane; arginine, a stimulator of exocytosis; and palmitate, β-oxidation and GPR40-dependent insulin secretion). *Cebpg* KD caused increased insulin secretion under all stimulatory conditions tested (Fig. 4L). This suggests that the effect of CEBPG is upstream of the mitochondria and exocytosis, and is in agreement with an effect of CEBPG on insulin production via UPR. To gain further understanding of the mechanisms of CEBPG-affected insulin secretion, we performed Ca²⁺ imaging, Seahorse, and metabolomic analysis of *Cebpg* KD INS-1

832/13 cells. KD of *Cebpg* in INS-1 832/13 cells had no effect on Ca²⁺ in response to glucose (Fig. S8C, D). Seahorse experiments (Fig. S8E-M) showed that *Cebpg* KD, in agreement with elevated ER stress, caused a moderate reduction in basal respiration, acute response, and proton leak (Fig. S8F, G, J). These observations are not in line with the observed enhanced insulin secretion after *Cebpg* KD (Figs. 3F and 4L). But our results using several insulin secretagogues (including the mitochondria fuel a-KIC) suggest that the effect of *Cebpg* KD is not primarily exerted at the mitochondrial level. Furthermore, metabolomic analysis revealed no effect of *Cebpg* KD on the 110 assessed metabolites, including key metabolites in central carbon metabolism, nor did it impact on changes in levels of these metabolites in response to glucose (Fig. S9). To gain further mechanistic insight into the role of CEBPG as a regulator of insulin secretion and insulin transcription, we analyzed *Cebpg* KD INS-1 832/13 cells with RNA-seq. 1433 genes were differentially expressed after FDR correction (Supplementary Data 5). GO-term analysis (Supplementary Data 5) using these genes confirmed a role for CEBPG in UPR ("negative regulation of UPR" upregulated), with *Ddit3, Herpud1, Atf3,* and *Hspa5* (gene for GRP78/BIP) being upregulated. Furthermore, both insulin genes (*Ins1, Ins2*) and two insulin transcription factor genes (*Mafb, Isl1*) were upregulated, thus providing a plausible mechanism for the effect of *Cebpg* KD on insulin transcription. In support of a role of CEBPG in beta cell growth regulation and in line with affected beta cell mass in CEBPG KO mice, GO-terms for cell cycle and apoptosis were identified (downregulated, including *Mki67*) (Supplementary Data 5). Thus, our data collectively imply that the effect of CEBPG on insulin secretion is explained by the effect of CEBPG on UPR and subsequent control of insulin production.

*TMEM176A/B* ("Microfilaments") are a pair of duplicated genes encoding a non-selective cation channel[51] reported to regulate cancer growth[52]. On a normal diet, *Tmem176a/b* knockout mice[53] were normoglycemic and showed unaffected insulin- and glucose responses during an intraperitoneal glucose tolerance test (Fig. S10A–D). However, *Tmem176a/b* knockout mice exhibited reduced beta cell mass (Fig. 5B) and islet size (Fig. 5C), in line with the reported role in cell growth. On a high-fat diet, *Tmem176a/b* knockout mice exhibited a negative acute insulin response (Fig. 5D, Fig. S10E) and perturbed glucose elimination (Fig. 5E) during an intraperitoneal glucose tolerance test. Furthermore, in agreement with the dGCNA prediction of altered beta cell microfilament organization, *Tmem176a/b* knockout mice (Fig. 5F, G) and *Tmem176a/b* KD INS-1 832/13 cells (Fig. 5H, I) had increased density of beta cell actin filaments, as assessed using phalloidin staining. In agreement, *Tmem176a/b* KD INS-1 832/13 had increased expression of the actin regulator *Cdc42*[54] (Fig. 5J). Having established that *Tmem176a/b* KD increases IBMX-induced, but not glucose-stimulated insulin secretion (Fig. 3F), we next tested the effect of *Tmem176a/b* KD on α-ketoisocaproic acid- and K⁺-stimulated insulin secretion in INS-1 832/13 cells. *Tmem176a/b* KD had no effect on insulin secretion under these conditions, suggesting that the effect is independent of mitochondrial metabolism and membrane depolarization (Fig. 5K). In agreement, Seahorse experiments showed that *Tmem176a/b* KD had no major impact on mitochondrial function (Fig. S10G–O) except for a minor reduction in basal respiration (Fig. S10H). Furthermore, *Tmem176a/b* KD had no effect on the 110 assessed metabolites, nor did it impact changes in levels of these metabolites in response to glucose (Fig. S9). The observation that *Tmem176a/b* KD affects insulin secretion only under conditions of elevated cAMP (with IBMX) speaks in favor of an effect of TMEM176A/B in the amplifying pathway of insulin secretion. cAMP is a mediator of the amplifying pathway and acts in part via direct effects on exocytotic proteins[55]. In agreement, the expression of genes encoding key exocytotic proteins (*Snap25, Syt11, Rab27a*[31,56]) was upregulated in *Tmem176a/b* KD INS-1 832/13 cells (Fig. 5L). Providing an explanation for increased *Ins1* expression after *Tmem176a/b* KD (Fig. 3D), *Nkx2.2*[37] expression was upregulated, whereas no effect on *Pdx1, Nkx6.1,* or *Neurod1* was

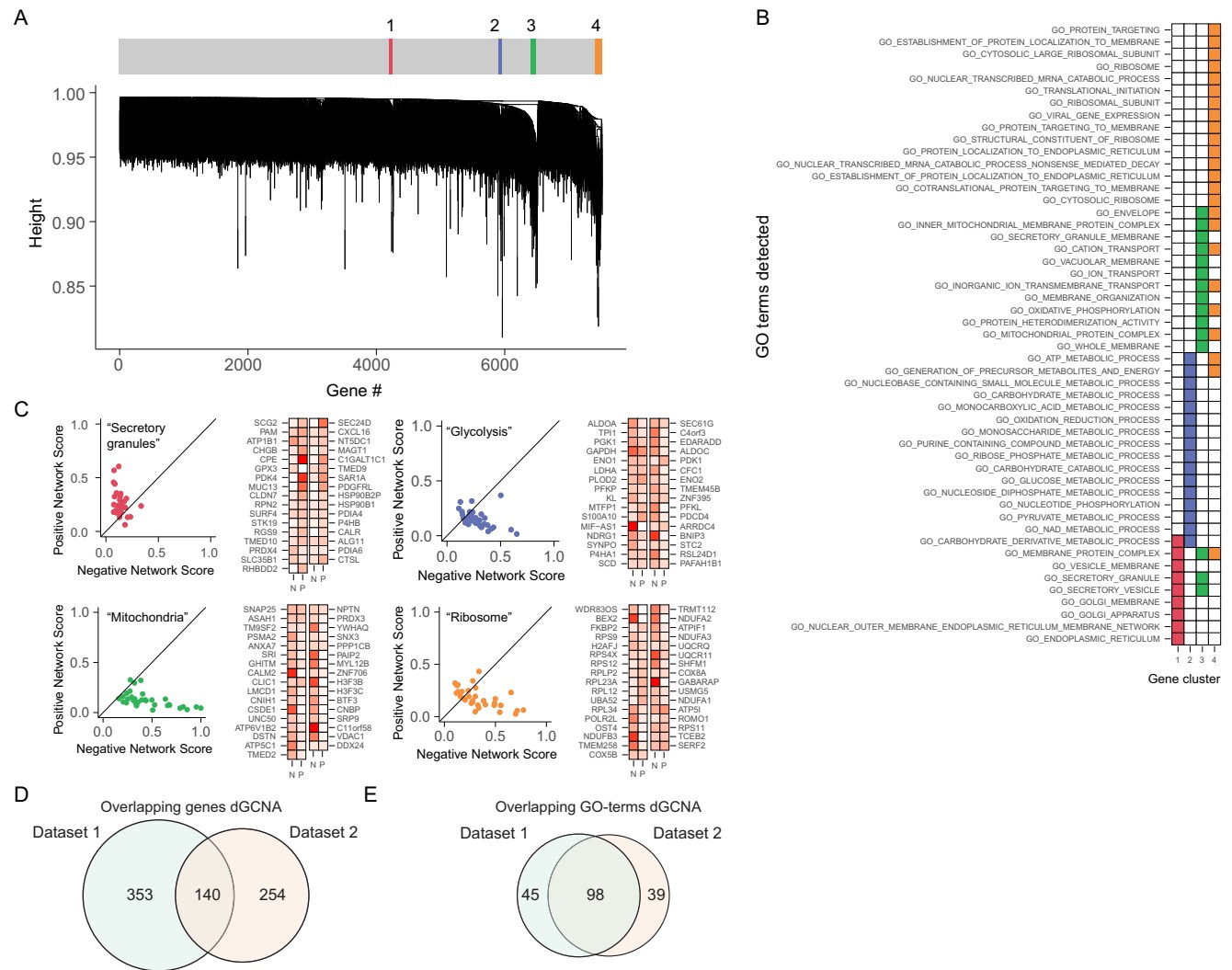

**Fig. 7 | dGCNA uncovers non-canonical T2D NDCGs in alpha cells. A** Dendrogram of four NDCGs in alpha cells. **B** GO-term (Biological process) enrichment is highly specific for each NDCG. Columns represent NDCGs and rows are top GO-terms. **C** Scatter plots representing coordination rank scores for genes at the tip of each NDCG, with negative (N) and positive (P) scores indicated. **D** Venn diagram of the overlap of genes in alpha cells between dataset 1 and dataset 2. **E** Venn diagram of the overlap of GO-terms for dGCNA genes in alpha cells between dataset 1 and dataset 2. See also Supplementary Data 6, Fig S4 and Fig. S11.

observed (Fig. 5M). Thus, we confirmed (as predicted by dGCNA) a role for TMEM176A/B in the regulation of microfilaments, which in turn affects insulin secretion.

To examine if T2D is associated with changed coordination in other islet cell types, we also analyzed alpha cells (Fig. 7). Using the merged dataset, we identified four alpha cell NDCGs (Fig. 7A) and named them according to biological functions (top GO-terms or function identified after literature review) (Fig. 7B and Supplementary Data 6). "Secretory granules" (hyper-coordinated) had GO-terms related to secretory granules, and contained *CHGB* and *SCG2*, as well as *CPE* and *PAM* (important for peptide hormone activation[57]) and several genes related to protein folding. "Glycolysis" (de-coordinated) contained multiple genes with key roles in glycolysis or hexose metabolism, including *ALDOA, TPI1, PGK1, GAPDH, ENO1, ENO2,* and *LDHA*. "Mitochondria" (de-coordinated) contained, e.g., *VDAC1, VDAC2*[58], *CALM2,* and *SRI*, suggesting a role also in Ca²⁺ handling[59]. Furthermore, the exocytosis mediator *SNAP25*[60] was present. "Ribosome" (de-coordinated) had strong GO-terms related to translation and contained multiple ribosome subunit genes (Fig. 7C, Supplementary Data 6). As in beta cells, GO-term analysis revealed a notable specificity for each NDCG (Fig. 7B). A comparison between alpha- and beta cell findings is

summarized in Fig. S11. Interestingly, 119/249 dGCNA genes were previously found to correlate with alpha cell exocytosis in a Patch-Seq study[61]. The highest enrichment was seen in the "Ribosome" NDCG with gene sets "Negative (1 mM) T2D" ($p = 1.56$e-14) and "Negative (10 mM) Healthy" ($p = 1.60$e-24) but also in "Mitochondria" and "Secretory Granules" across gene sets (Fig. S4B and Supplementary Data 3). Furthermore, "Glycolysis" showed enrichment for GWAS genes associated with fasting glucose (p-adj = 0.022) (Fig. S3C and Supplementary Data 3). As in beta cells, dGCNA results in alpha cells were overlapping between dataset 1 and dataset 2 for genes (Fig. 7D) and GO-terms (Fig. 7E). DEseq2 identified only 16 differentially expressed genes in alpha cells (Supplementary Data 6).

## Discussion

Here, we introduce dGCNA as a tool to interrogate scRNAseq data to reveal T2D-associated differential coordination of gene expression in islet cells, thereby inferring altered cellular functions. In addition to pathways and genes experimentally shown to underlie beta cell dysfunction, we identified non-canonical pathways and multiple genes with unknown roles in beta- and alpha cell function. The ability of dGCNA to predict the functional context of T2D-affected beta cell

genes was validated in both in vitro experiments and in mouse models. Furthermore, we replicated our findings between two datasets and showed that dGCNA outperforms differential expression analysis for the identification of T2D-associated changes in gene expression across datasets.

Using dGCNA, we identified eleven NDCGs affected in T2D beta cells. Most NDCGs contained multiple genes with established roles in beta cell function (including several GWAS genes), as well as a multitude of genes associated with T2D. A majority of the NDCGs were de-coordinated, which we interpret as a perturbed function. In line with a body of evidence[39,62], we observed mitochondrial dysregulation indicated by a de-coordinated NDCG containing genes encoding for complex 1 and 4 of the electron transport chain. Also, in support of perturbed coordination of stimulus-secretion coupling, we identified a de-coordinated NDCG containing regulators of glycolysis and a de-coordinated NDCG ("Glucose response") with a high density of key regulators of beta cell function, including *GCK* (the rate-limiting enzyme in glucose metabolism[63]), suggesting perturbed glucose sensing in T2D beta cells.

Two NDCGs ("Proliferation" and "Microtubuli") adjacent to "Glucose response" were also de-coordinated. These three NDCGs together contained virtually all known beta cell transcription factors, suggesting a general perturbation of beta cell transcriptional identity and reduced beta cell proliferation in T2D. "Proliferation" contained *PAX6*[34], *ISL1*[28], and several genes known to regulate proliferation. Beta cell mass is controlled by a balance between apoptosis and renewal of beta cells[64]. Most studies agree that beta cell mass is reduced in human T2D, although its contribution to insulin deficiency has been questioned[1,65,66]. Furthermore, "Microtubuli" containing *NKX2.2*[37] and genes related to the regulation of microtubules were de-coordinated. In further support of altered cytoskeletal arrangement in T2D beta cells, "Microfilaments", containing multiple genes related to actin filament arrangement, was also de-coordinated. Phalloidin staining of human pancreatic sections showed increased beta cell actin density in T2D donors, possibly representing a malfunctioning control of filament organization. The role of actin in relation to insulin secretion is two-fold. Firstly, it acts as a physical barrier impeding the access of insulin granules to the cell periphery. Secondly, through glucose-induced remodeling, it acts as a cytoskeletal structure for the transport of insulin granules to the plasma membrane[67,68]. *TMEM176A/B* in the "Microfilaments" network was identified as a regulator of insulin production and secretion in cell lines and in a mouse model. Confirming a role for TMEM176A/B in regulation of microfilament organization, *Tmem176a/b* KO mouse beta cells had increased density of actin, reminiscent of our data in T2D donors, and in agreement with the observed hampered insulin secretory response. In INS-1 832/13 cells, the *Tmem176a/b* KD-induced increase in actin density was paralleled by increased expression of *Cdc42*, a key regulator of actin dynamics[54]. Furthermore, *TMEM176* genes were recently shown to regulate Akt/mTOR-dependent pathways that are important regulators of actin remodulation[69]. As in the *Tmem176a/b* knockout mice, *Tmem176a/b* KD in INS-1 832/13 cells affected insulin secretion; however, for reasons that we do not yet understand, the effect was opposite and restricted to cAMP-enhanced insulin secretion. cAMP stimulates insulin exocytosis via a direct effect on exocytosis proteins that are highly dependent on actin fiber organization[55]. In agreement, *Tmem176a/b* KD caused increased expression of three mediators of insulin exocytosis (*Snap25*, *Syt11*, and *Rab27a*)[31,56]. Thus, our functional validation confirmed the dGCNA prediction of a role for TMEM176A/B in microfilament organization and suggests that the effect of TMEM176A/B on insulin secretion is mediated via its effect on actin filament organization and subsequent effects on exocytotic proteins.

Moreover, we found a de-coordinated network of UPR genes. This indicates a perturbed capacity to remove misfolded proteins that could be contributing to beta cell failure in T2D. A central gene in "UPR" was *CEBPG*, a redox regulator that heterodimerizes with *ATF4*[48].

Our functional validation in a mouse model, cell lines, and human islets confirmed the dGCNA prediction and put forward *CEPBG* as an important regulator of all three arms of the UPR pathway in beta cells. Targeting of CEBPG resulted in failed upregulation of UPR regulators and ensuing accumulation of aggregated proteins, as well as increased insulin transcription. Our data suggest that the effect of CEBPG on insulin secretion is related to its effect on UPR and subsequent effect on insulin production. It needs to be mentioned that both the *Tmem176a/b* KO mice and the *Cebpg* KO mice were global KOs, and potential effects in non-beta cells could be contributing to an insulin-related phenotype. However, our confirmation of a functional role of these genes in beta cell lines suggests that they have important functions in beta cells.

The de-coordinated processes likely represent pathophysiological mechanisms of T2D in beta cells. Conversely, we observed hyper-coordination in three NDCGs. "Ribosomes" were composed of ribosome subunit genes, indicative of increased translation of insulin. Importantly, we confirmed increased immunoreactivity for the ribosome marker RPS3 in islets of T2D patients. "Insulin secretion" was composed of genes related to the exocytosis machinery, insulin processing, and insulin granule content. Interestingly, a hyper-coordinated NDCG containing genes related to lysosomes was identified close to "Insulin secretion". Lysosomes degrade insulin granules, via autophagy or crinophagy, and play an important role in protecting the beta cell in situations of stress[70]. The observation that this NDCG was devoid of autophagy genes speaks in favor of upregulated lysosomal degradation of insulin granules via crinophagy in T2D beta cells. The "Lysosome" NDCG also contained the lysosome-associated gene Inceptor/*KIAA1324*, recently identified in mice and human cell lines as a regulator of beta cell insulin sensitivity[33], confirming its altered regulation also in T2D patients.

We also performed dGCNA on alpha cell data and identified four NDCGs affected in T2D alpha cells. It should be mentioned that, in accordance with other FACS-based studies[15], alpha cells were over-represented in our datasets. T2D patients commonly fail to suppress glucagon levels in the postprandial state. This leads to increased hepatic glucose production and aggravated hyperglycemia[71]. As in beta cells, T2D alpha cells displayed a de-coordinated network of glycolysis genes. Our data gain support from previous reports on lower expression of glycolysis genes in alpha cells of T2D donors[72] and in Gad[+] non-diabetic donors[73]. Mice lacking the rate-limiting enzyme (*Gck*) of glycolysis in alpha cells were recently shown to have hampered glucose-induced suppression of glucagon secretion[63]. Furthermore, a hyper-coordinated network with genes related to secretory granules and peptide hormone processing was evident. Disrupted glucose-induced suppression of alpha cell exocytosis of secretory granules has been elegantly demonstrated in T2D alpha cells[61]. In fact, multiple alpha cell dGCNA genes were overlapping with the alpha cell exocytosis-related genes identified in ref. 61. Finally, as in beta cells, we identified a de-coordinated network containing mitochondrial genes. Mitochondrial function is necessary for glucose-induced suppression of alpha cell exocytosis[61], and hyperglycemia induces alpha cell mitochondrial dysfunction[74]. Thus, our data on perturbed glycolysis, affected secretory granule function, and perturbed mitochondrial function are in line with previously observed malfunctioning glucose-induced suppression of glucagon secretion in T2D and provide further insight into the underpinning gene regulation. Contrary to beta cells, in T2D alpha cells a de-coordinated network composed of ribosome subunit genes was evident. This suggests reduced glucagon translation and could be interpreted as a compensatory mechanism, trying to lower glucagon secretion and thereby glucose output from the liver. Furthermore, in contrast to the beta cells, the alpha cells did not display any alterations in UPR or cytoskeleton-related gene networks. Thus, our findings confirm and expand previous knowledge on alpha cell alterations in T2D and highlight cell type-specific alterations in T2D.

An important limitation for the dGCNA pipeline is the read depth required for the full analysis. The two datasets presented consisted of ~1 M reads/cell, which allowed for the analysis. Typical read depth (15k–150k reads/cell) from methods that prioritize the number of cells rather than information per cell does not provide enough signal for a successful topological analysis.

By harnessing the individual variation between cells within a cell type, the dGCNA algorithm identified cell type-specific, coordinated regulatory changes in T2D with unprecedented power. Our analysis generated an atlas of molecular changes in distinct populations of islet cells and thus allows for a bird's-eye view of disease-related changes, including multiple targets for future validation. Our findings support a model where basal insulin production and the capability of exocytosis are enhanced in T2D beta cells, but the stimulus-secretion coupling of insulin release and degradation of peptides that failed quality control are perturbed. This is in line with current clinical interventions based on enhancing insulin secretion using, e.g., sulfonylurea or GLP-1-based approaches[30]. In addition, our findings suggest that T2D beta cells are less differentiated, less capable of adaptive growth, display cytoskeletal remodeling compatible with perturbed insulin secretion, and show signs of enhanced lysosome activity. Furthermore, our data points to cell type-specific alterations with non-overlapping programs in beta- and alpha cells.

While we assumed cell-to-cell variability, the coordination of biological programs reported herein is unbiased: we have listed all networks detected, each of which was highly specific with respect to GO-terms, supporting the hypothesis that the changes in gene coordination were indeed coordinated on the level of pathways. Our analysis, without given prior information, revealed dysregulated canonical pathways and multiple genes with experimentally proven importance in T2D beta cell dysfunction. It also revealed non-canonical processes and multiple genes associated with T2D pathophysiology. Notably, the expression level (mRNA or protein) did not need to be altered for a gene to be identified as disease-regulated. The increased power of our analysis could potentially have a great impact on the future design of perturbation experiments as well as the development of pharmacological agents targeting the newly identified pathways or genes in future studies. Although we have not measured a temporal aspect of the gene coordination, we hypothesize that it represents changes in the synchronization of gene expression. Interestingly, such a theory would entail that separate biological pathways are controlled with temporal specificity in cells in general. Our approach holds promise for use in elucidation of disease mechanisms of other complex tissues and diseases, and warrants studies of the possibility of temporal coordination of gene expression.

## Methods

### Human islet donors
Information about the donors included in the study is presented in Supplementary Data 1. Human islets for dataset 1 were obtained from the Nordic Network for Islet Transplantation (www.nordicislets.org). Written informed consent was obtained from the donors or their relatives, and the ethics committees at Uppsala and Lund Universities approved all procedures. Human islets from dataset 2 were purchased from Prodo Laboratories Inc. (Irvine, CA, USA), providing pancreatic islets isolated from donors obtained with research consent from Organ Procurement Organizations (OPOs)[75]. The use and storage of human islets and tissue samples were performed in compliance with the Declaration of Helsinki, ICH/Good Clinical Practice, and were approved by the Regional Ethics Committee (Gothenburg, Sweden).

### Sample processing and single-cell RNA sequencing
Human islets were stored in 30 ml HEPES-buffered medium. Islets were transferred to a 50 ml tube and centrifuged at 150 g for 2 min at RT. Supernatant was removed, and cells were washed once with 5 ml of Accutase. Another 5 ml of Accutase was added for incubation at 37 °C for 8–10 min. Thereafter, 5 ml of cold islet media was added, and cells were suspended by pipetting. Cells were passed through a 40-µm cell strainer and centrifuged at 500 × g for 5 min and washed twice with PBS at 500 × g for 5 min. Finally, cells were resuspended in HBSS and kept on ice before being FACS-sorted as single cells into 384-well plates. cDNA libraries were generated using the Smart-seq2 protocol[27] and sequenced on a HiSeq 2500 sequencer.

### Filtering and cell type clustering
For dataset 1. STAR 2.4.2a was used for alignments with the reference genome hg38 using 2-pass alignment for improved performance of de novo splice junction reads, filtered for uniquely mapping reads only, and saved in .bam files. The count matrix showed the individual counts aligning to each gene per cell. Expression values were computed as reads per kilobase of the gene model and million mappable reads (RPKMs). Sequenced cells with mRNA reads <100,000, percent of uniquely mapping reads <50%, or percent of uniquely exonic reads <40% were removed, obtaining 3645 transcriptomes.

For dataset 2, raw sequencing data of human pancreatic islets were demultiplexed and converted into fastq files using Illumina bcl2fastq with default settings. Reads from human data were aligned to the human genome hg38 using the STAR aligner. Uniquely aligned reads mapping to the RefSeq gene annotations were used for gene expression estimation at reads per kilobase transcript and million mapped reads (RPKMs) using rpkmforgenes. Low-quality cells were excluded from downstream analysis when they failed to meet the following criteria for retaining cells: (1) ≥50,000 sequence reads; (2) ≥40% of reads uniquely aligned to the genome; (3) ≥40% of these reads mapping to RefSeq annotated exons; and (4) ≥1000 genes with RPKM > = 1. In addition, doublets detected by Scrublet were further removed, resulting in a total of 4866 single cells for downstream analysis. For cell type assignment, the top 1000 most variable genes were used for PCA dimension reduction, followed by clustering using affinity propagation, resulting in seven non-endocrine cell clusters (acinar, ductal, PSC, MHC II, mast, stem-like, endothelial cells) and one endocrine cell cluster. To further annotate endocrine cell types, we applied the same approach to endocrine cells only and identified five clusters representing alpha cells, beta cells, delta cells, gamma cells, and epsilon cells, respectively. For the number of cells in each cluster and dataset by disease status, see Supplementary Data 1.

### Integration of datasets
For datasets 1 and 2, the expression of transcript variants with the same gene name was summed. The datasets were integrated and clustered using Conos, and cell types were assigned using label propagation based on previous clustering analysis and known markers. We used the basicP2proc function from Pagoda to preprocess each dataset and facilitate integration. A conos object was created with 'Conos::new()' to analyze the joint dataset. Clustering and dimensional reduction were performed using the buildGraph function from pagoda. A shared nearest neighbor (SNN) graph was built using principal component analysis (PCA) as the dimensionality reduction method, retaining the top 40 components. Global clusters were detected using the Leiden clustering algorithm implemented in the findCommunities function from Pagoda. The graph was embedded into a two-dimensional space using UMAP for visualization, and markers were selected based on a combination of differential expression and biological expertise.

### Identification of cell type-specific gene expression
For Fig. 1C, we plotted canonical markers for each cell type in the integrated dataset (1 + 2). For Fig. 1D, we ran "getDifferentialGenes" function in Conos and plotted the top 5 Z-scores ordered by the specificity of expression.

## Differential Gene Coordination Network Analysis (dGCNA)

Cell type-specific gene-gene correlation matrices for each cell type with more than 30 cells/donor were constructed separately for T2D and non-T2D donors using Pearson correlation between transcriptional levels in $log2(RPKM+1)$ of each gene pair. We filtered non-expressed genes ($RPKM+1 < 2$) and genes detected ($RPKM+1 > 2$) in less than 5 cells/donor. The correlations were calculated using a linear mixed-effects model including random effects to correct for donor-specific expression differences. To build cell type-specific differential networks between disease and healthy states, we calculated the difference in pairwise Pearson correlations for each gene pair between T2D and non-T2D donors. To remove false positive weights in the differential network, we used a bootstrapping approach, generating random differential networks by selecting 512 pairs of random groups of donors. We filtered all correlations with a relative frequency lower than 0.975, keeping only links that were significantly stronger than random. This approach provides an internally scaled thresholding that takes the quality of the underlying dataset into account. We then performed clustering of the differential networks (one per cell type) using *TOMdist* (WGCNA package) and hierarchical clustering (*hclust*) to reveal gene communities altered between the conditions. Networks of differentially coordinated genes (NDCGs), i.e., gene modules, were determined using the *cutreeDynamic* function (WGCNA package) at different thresholds. Next, we evaluated whether these T2D-related NDCGs were associated with known biological functions using a hypergeometric test (Bonferroni correction, $p < 0.05$) with Gene Ontology terms (Cellular component, Biological function, and Molecular function). Protein-coding genes detected in our dataset for each cell type were used as the background gene-set. Network scores were calculated with the function *eigen_centrality* (igraph R package) for the positive (hyper-coordinated) subnetwork (network with just positive weights) and for the negative (de-coordinated) subnetwork (network with just negative weights) separately.

## Differential expression analysis

We used the DESeq2 method to calculate the differential expression of genes. Expression of all single-cell counts of the same cell type and the same donor was summed before calculating differential expression. We removed unwanted variation from the RNA-seq counts with the package RUVSeq using control genes. Thresholds of the volcano plots were established to a fold change of 2 and an adjusted *p*-value of 0.05. For bulk RNA-Seq data, the expression count matrix was generated with Salmon v1.5.0, and differential expression analysis was performed with DESeq2 v1.30.1.

## Evaluating the replicability of the dCGNA analysis

We evaluated similarity between the picked modules in datasets 1 and 2 by calculating the pairwise Jaccard index, considering the overlap in significant GO-terms.

## Enrichment analysis of published datasets (Patch-Seq, GWAS, and whole genome CRISPR screen)

We used a hypergeometric test for enrichment of overlap between the dCGNA modules in T2D- and islet-related gene sets. To decrease the loss of power due to multiple comparison correction, we required a minimum overlap size of 2 or more genes for a test to be performed. All expressed genes in the cell type corresponding to the module were considered as background. We computed the adjusted *p*-value within each gene-set for all modules using the FDR method.

## Selection of genes for functional validation

We selected genes for functional follow-up based on the following criteria: high eigencentrality in the RDN from different modules. We

then applied filters: (1) no previous publication with search terms "beta cell" or "islet" on PubMed, (2) robust expression in INS-1 832/13 cells (assessed with bulkRNAseq). One exception was made for *DBI*, which had two older papers in a hamster cell line and rat islets[76,77]. *TMEM176A/B* and *CEBPG* were selected for further in vivo studies due to the availability of KO mouse models.

## INS-1 832/13 cell culture and siRNA-mediated gene silencing

INS-1 832/13 cells[47] were cultured at 5% $CO_2$ and 37 °C in RPMI1640 medium (Sigma Aldrich, St Louis, MO) containing 2 g/L D-glucose, supplemented with 10% fetal bovine serum, 10 mM HEPES, 1 mM sodium pyruvate, and 50 μM β-mercaptoethanol (Sigma Aldrich). Gene silencing in INS-1 832/13 cells was performed using Lipofectamine RNAiMAX (Life Technologies, Waltham, MA) and 60 nM siRNA targeting *Atraid* mRNA (s151757 and s235878), *Cebpg* mRNA (s129415 and s129417), *Dbi* mRNA (s128869 and s128871), *Dstn* mRNA (282408 and 282409), *Gabarap mRNA* (s133040 and s133041), *Spire1* (s158071 and s158073), *Tmem176a* mRNA (s150574) and *Tmem176b* mRNA (s140062). The sequences for scrambled siRNA were sense: 5´-GAGACCCUAUCC-GUGAUUAtt-3´ and antisense: 5´-UAAUCACGGAUAGGGUCUCtt-3´ (Silencer Select, rat negative control #1; scrambled siRNA and all siRNAs were from Ambion, Life Technologies). Transfection complexes were prepared in accordance with the instructions provided by the manufacturer and added to $2 \times 10^5$ cells seeded per well in 24-well plates.

Human islets were transfected with 60 nM siRNA targeting human *CEBPG* (s2901, Silencer Select Pre-designed siRNA, Ambion, Life Technologies). The sequence for *CEBPG* siRNA was sense: 5´-AGAGCCGGUUGAAAAGCAAtt-3´ and antisense: 5´-UUGCUUUU-CAACCGGCUCUtt-3´.

## RNA extraction

Total RNA was isolated from INS-1 832/13 cells 48 h after transfection using the NucleoSpin extraction kit (Macherey-Nagel, Düren, Germany). The amount of isolated RNA was measured using the NanoDrop system. For bulk RNA sequencing, samples were also analyzed using TapeStation. Total RNA from human islets was extracted using Nucleospin RNA XS (Macherey-Nagel) three days post-transfection.

## cDNA synthesis and real-time qPCR

For the experiment, 1 μg of isolated total RNA was reverse-transcribed to cDNA using the RevertAid First Strand cDNA synthesis kit (Life Technologies). qPCR was performed with 25 ng cDNA template using TaqMan Expression Master Mix (Life Technologies) according to the instructions provided by the manufacturer. *Tbp* and *Hprt1* (Rn01527840_m1 and Rn01455646_m1, respectively) were used as housekeeping genes. Expression levels were calculated using the $2^{-\Delta\Delta Ct}$-method. TaqMan assays used were: *Atf4* (Rn00824644_g1), *Atf6* (Rn01490844_m1), *Atraid* (Rn01468747_g1), *Cebpg* (Rn01764319_m1), *Dbi* (Rn00821402_g1), *Dstn* (Rn01415640_g1), *Gabarap* (Rn00490680_g1), *Ins1* (Rn0212433_g1), *Ins2* (Rn01774648_g1), *Spire1* (Rn01491339_m1), *Tmem176a* (Rn01451723_m1), *Tmem176b* (Rn00508100_m1), *Xbp1s* (Rn03464499), *Cdc42* (Rn00696671), *Nkx2.2* (Rn04244749), *Nkx6.1* (Rn01450076), *NeuroD1* (Rn00824571), *Pdx1* (Rn00755591), *Rab3a* (Rn07311159), *Rab27a* (Rn00568995), *Snap25* (Rn00578534), *Syt11* (Rn01218287), and *Xbp1* (Rn01443523_m1). For human islets, 300 ng of RNA was reverse-transcribed to cDNA using RevertAid First Strand cDNA synthesis kit. RT-qPCR for target genes and two endogenous controls (*HPRT1* and *TBP*) was performed using 2.5 ng cDNA. TaqMans for human islet were: *ATF4* (Hs00909569_g1), *ATF6* (Hs00232586_m1), *CEBPG* (Hs01922818_s1), *DDIT3* (Hs99999172_m1), *HPRT1* (Hs4326321_m1), *INS* (Hs02741908_m1), *TBP* (Hs00427620_m1), and *XBP1* (Hs00231936_m1). All TaqMan assays were from Life Technologies, and qPCR reactions were run on the Viia7 real-time PCR system (Applied Biosystems, Foster City, CA).

## Insulin secretion experiments

INS-1 832/13 cells were seeded in 24-well plates ($2 \times 10^5$ cells/well) and allowed to adhere for approximately 5 h before transfection. On the day of insulin secretion, cells were washed in PBS, incubated with 2.8 mM glucose for 2 h, and then incubated for 60 min with 2.8 mM glucose, 16.7 mM glucose, 16.7 mM glucose +0.1 mM IBMX, 2.8 mM glucose +35 mM $K^+$, 2.8 mM glucose +10 mM α-ketoisocaproic acid, 16.7 mM glucose +10 mM arginine, or 16.7 mM glucose +1 mM palmitate. For the experiments, glucose was dissolved in the secretion assay buffer (SAB; 114 mM NaCl, 4.7 mM KCl, 1.2 mM $KH_2PO_4$, 1.16 mM $MgSO_4$, 1.16 mM $CaCl_2$, 20 mM HEPES, 25.5 mM $NaHCO_3$, and 0.2% BSA). The cells were then placed on ice, and the culture media were collected for insulin and protein determinations. Insulin concentration was determined by ELISA from Mercodia (Uppsala, Sweden), and protein concentration was determined using Protein Assay Dye Reagent (BioRad, Hercules, CA).

## Thapsigargin treatment

INS-1 832/13 cells were seeded in 24-well plates ($2 \times 10^5$ cells/well). Cells were incubated with 100 nM thapsigargin (Sigma Aldrich) for 24 h. In a second experiment, cells were seeded in 24-well plates and *Cebpg* mRNA was silenced using siRNA (s129415) for 24 h. 100 nM thapsigargin was then added to these cells for 24 h. Total RNA was extracted from the cells from both above-mentioned experiments.

## Immunohistochemistry

Primary antibodies: anti-rabbit ATF6 (code NBP1-76675, dilution 1:200, Novus Biologicals, Centennial, CO), anti-rabbit CEBPG (code HPA012024, dilution 1:20, Sigma Aldrich), anti-rabbit DSTN (code ab186754, dilution 1:100, Abcam), anti-rabbit GABARAP (code HPA-78365, 1:500 dilution, Sigma Aldrich), anti-guinea pig insulin (code A0564, 1:1000 dilution, Dako, Glostrup, Denmark), anti-rabbit KIAA1324 (code PA5-67123, dilution 1:800, Life Technologies), anti-rabbit SPIRE1 (ab130403, 1:1000 dilution, Abcam), anti-rabbit TMEM176A (code NBP1-83283, 1:20 dilution, Novus Biologicals), anti-rabbit TMEM176B (code CSB-PA023758LA01HU, 1:50 dilution, Cusa-Bio, Houston, TX), anti-rabbit DBI (code DBI-AK3, 1:800 dilution (kind gift from Prof. Efendik, Karoliska Institute, Stockholm, Sweden)), anti-rabbit ATRAID (code bs-9806R, 1:200 dilution, Thermo Scientific), and anti-rabbit RPS3 (code ab128995, 1:200 dilution, Abcam).

All antibodies were diluted in PBS with 0.25% BSA and 0.25% Triton X-100. Slides were incubated with primary antibodies overnight at 4 °C, then washed twice in PBS with 0.25% Triton X-100. Slides were then incubated for 1 h with secondary antibodies at RT. Secondary antibodies used: Donkey anti-guinea pig AlexaFluor594 (for insulin), donkey anti-rabbit Cy2 (for ATF6, CEBPG, DSTN, GABARAP, KIAA1324, SPIRE1, TMEM176A, TMEM176B, DBI, ATRAID, RPS3), and goat anti-guinea pig AlexaFluor405 (for insulin). For phalloidin staining, Phalloidin-iFluor488 (Abcam) was diluted in PBS (+0.25% Triton X-100 and 0.25% BSA; 1:1000) and incubated for 60 min at RT. Slides were then washed twice in PBS with 0.25% Triton X-100 and mounted. Immunofluorescence was examined in an epifluorescence microscope (Olympus BX60, Olympus, Tokyo, Japan) or a Zeiss LSM800 confocal microscope (Oberkochen, Germany). Images were acquired with a digital camera (Olympus DP74, Olympus) using the CellSens Dimensions software (Olympus) or Zen System 2.6 (Zeiss). Pixel intensity was measured using ImageJ software, and colocalization (expressed as Pearson´s colocalization coefficient) analysis was performed using the CellSens Dimensions software in a blinded manner.

## Animals

*Tmem176a/b* double knockout (DKO) mice have been described in detail elsewhere[53]. For this study, the mice were kept in Macrolon cages in a temperature-controlled environment (21 °C) with a relative humidity of 40–60%, on a 12 h light/dark cycle and with free access to

diet (3 weeks of standard rodent chow or high-fat diet (60% fat, 20% protein, 20% carbohydrates, Special Diet Services, Witham, UK)) and tap water. All experiments were performed in accordance with institutional guidelines for animal care and use for the University of Nantes. The *Cebpg* KO mice[48] were kept in Macrolon cages in a temperature-controlled environment (21 °C) with a relative humidity of 40–60%, on a 12 h light/dark cycle, and with free access to standard rodent chow and tap water. All animal experiments were approved by the animal ethics committee in Malmö and Lund, Sweden.

## Beta cell mass measurements in *Cebpg* KO mice and *Tmem176a/b* DKO mice

Male *Tmem176a/b* DKO and WT ($n = 8$ per group, age: 12–14 weeks, weight: 28.5 + 3) and *Cebpg* KO ($n = 6$) and WT ($n = 16$) (50% males, age: 12 weeks) mice were analyzed. Beta cell mass, average islet size, and total islet number measurements in the KO mouse models were performed as detailed[78] at three depths of the pancreas (100 μm difference between a series of three sections). Insulin was visualized using an anti-insulin antibody (code A0564, 1:1000 dilution, Dako) and anti-guinea pig AlexaFluor594 (1:400 dilution), and the signal was used to determine the total beta cell area, which was divided by the total pancreas area to generate a measure of beta cell mass. Immunofluorescence was examined under an epifluorescence microscope (Olympus BX60, Olympus, Tokyo, Japan) with a digital camera (Olympus DP74, Olympus). Measurements were made at 40× magnification using the CellSens Dimensions software (Olympus) in a blinded manner.

## Intraperitoneal glucose tolerance test (IpGTT)

Male *Tmem176a/b* DKO mice fed HFD ($n = 10$, age: 15 weeks, weight: 28 + 2.7 g) or a control diet ($n = 10$, age 12 weeks, weight: 25 + 2.9) and male WT mice fed HFD ($n = 10$, age 14 weeks, weight: 27 + 0.9) or a control diet ($n = 10$, age 13 weeks, weight: 24.6 + 2.7) were used, as previous studies show that males develop more pronounced insulin resistance and beta cell stress compared with females[79]. After a 4-h fasting period, basal blood samples were drawn via retro-orbital puncture. The mice were then injected intraperitoneally with glucose (2 g/kg bodyweight). Blood samples were collected after 10, 20, 30, 60, and 90 min into chilled EDTA-tubes. Samples were maintained on ice and centrifuged at $500 \times g$ for 5 min, and plasma was stored at −80 °C until analysis.

## Insulin secretion and glucose measurement

Plasma insulin from the *Tmem176a/b* DKO mice subjected to IpGTT and fasting plasma insulin from the *Cebpg* KO mice were determined by ELISA (Mercodia) according to the manufacturer's instructions. Glucose from mouse plasma was analyzed using the Infinity Glucose (Ox) kit from Thermo Scientific according to the instructions provided by the manufacturer. Acute insulin response (AIR) in the *Tmem176a/b* DKO mice was calculated as the difference between 10-min insulin levels and basal insulin levels. Integrated area under the curve (iAUC) for the glucose levels in the *Tmem176a/b* DKO mice subjected to IpGTT was calculated using GraphPad Prism 8 (GraphPad Prism, San Diego, CA).

## Mitochondrial oxygen consumption measurements

Oxygen consumption rates (OCRs) were measured by Seahorse XFe24 Extracellular Flux Analyzer (Agilent Technologies, USA). INS-1 832/13 cells (120,000 cells/well) in poly-L-lysine-coated Seahorse XF cell culture plates were preincubated in a $CO_2$-free incubator for 2 h at 37 °C in assay buffer (114 mM NaCl, 4.7 mM KCl, 1.2 mM $KH_2PO_4$, 1.16 mM $MgSO_4$, 20 mM HEPES, and 2.5 mM $CaCl_2$, pH 7.2) with 2.8 mM glucose. To determine changes in mitochondrial respiratory response, stabilized cellular respiratory rate was measured before and after injection of 16.7 mM glucose and subsequently after sequential injection of 4 μg/

ml oligomycin, 2 μM FCCP, and 1 μM rotenone. Aside from injection timings, experiments were performed as previously described[80]. Primary data analysis was performed using the Seahorse Analytics web tool (https://seahorseanalytics.agilent.com/; version 1.0.0-699).

## Metabolomic analysis

Metabolites were extracted and profiled using gas chromatography/mass spectroscopy, as previously described[81]. Metabolite data were mean-centered and scaled, followed by analysis by principal component analysis (PCA; *prcomp*) and linear models (*lm*; *Anova*, car R package). Plots were produced in *ggplot* (ggplot2 R package). The analysis provided 623 features, out of which 110 were putatively identified by mass spectrum and retention index using the MS-DIAL associated libraries and in-house libraries.

## Ca²⁺ imaging

INS-1 832/13 cells were cultured in imaging-specific plates coated with poly-L-lysine (1μg/ml working solution) from Nunc (Lab-Tek Chambered #1.0 Borosilicate Coverglass System) in regular RPMI1640 media (Sigma Aldrich). $Ca^{2+}$ measurements were performed as detailed[82] 72 h after transfection. Briefly, cells were washed in PBS and then pre-incubated in 2.8 mM glucose for 2 h at 37 °C. 1.5 h into the incubation, cells were loaded with 2.5 μM, 0.5 mM sulfinpyrazone and incubated for the remaining period. Prior to imaging, the pre-incubation buffer was discarded, and the cells were washed with imaging buffer. Upon imaging, a 2-min baseline (2.8 mM glucose) was recorded before the addition of a highglucose (16.7 mM) buffer. Images were analyzed using ImageJ software.

## Western blotting for Grp78/BiP

INS-1 832/13 cells were lysed in RIPA buffer 72 h after transfection. Protein content was measured using the BCA Protein assay kit from Pierce. 20 μg of protein was loaded onto Mini PROTEAN TGX stain-free gels (BioRad). Precision Plus Protein Kaleidoscope Prestained Protein Standards (#1610375, BioRad) were used. Gels were activated using a ChemiDoc MP CCD camera from BioRad. Blotting was performed using a Transblot Turbo Transfer System (BioRad). Membranes were activated and then blocked for 1 h using TBS-T with 5% non-fat dry milk. Then, membranes were incubated overnight at 4 °C with Grp78 primary antibodies (#3183S, Cell Signalling, Danvers, MA), diluted 1:1000 in TBS-T with 5% non-fat dry milk. Membranes were then washed three times in TBS-T, incubated with secondary antibody diluted in TBS-T with non-fat dry milk for 60 min, washed three times in TBS-T, developed using SuperSignal West Pico Maximum Sensitivity Substrate (Thermo Scientific), and imaged using a ChemiDoc MP CCD camera. Proteins were quantified based on total protein normalization with stain-free gels in Image Lab version 6.0.1 (BioRad).

## Protein aggregation assay

Thioflavin T (ThT) is a benzothiazole dye that exhibits enhanced fluorescence when binding to ER stress-induced protein aggregates[83]. ThT was dissolved in 70% ethanol (60 mM) and stored at −20 °C. Immediately before use, ThT was diluted in RPMI1640 to create a 5 mM stock solution. The stock solution was further diluted in assay media to a final concentration of 5 μM. INS-1 832/13 cells were seeded into 96-well plates at a total density of $1 \times 10^4$ cells/well for *Cebpg* KD. Then, 72 h post-transfection, cells were treated with thapsigargin (500 nM) and ThT (5 μM), and fluorescence (excitation 437 nm, emission 485 nm) was read after a 1-h incubation at 37 °C.

## Phalloidin staining in INS-1 832-13 cells

Cells were transfected with *Tmem176a* and *Tmem176b* siRNAs in 8-well chamber slides (Sarstedt). 72 h after transfection, cells were incubated with anti-guinea pig insulin antibodies (Dako) overnight at 4 °C, washed twice in PBS (with 0.25% BSA and 0.25% Triton X-100),

incubated with donkey anti-guinea pig AlexaFluor594 (Jackson ImmunoResearch) for 60 min at room temperature, again washed twice with PBS (with 0.25% BSA and 0.25% Triton X-100). Phalloidin (iFluor488 from Abcam) was diluted 1:1000, and cells were stained for 60 min at RT. Phalloidin intensity was quantified using ImageJ software in a blinded manner.

## Analysis and statistics

All experimental data were analyzed using 2-way ANOVA, followed by Tukey's test post hoc, or by two-tailed Student's *t*-test. Differences with $p < 0.05$ were considered significant. All replicates mentioned were biological replicates.

## Reporting summary

Further information on research design is available in the Nature Portfolio Reporting Summary linked to this article.

## Data availability

The newly generated human islet scRNAseq datasets were deposited at GEO and EBI on the following accession numbers: GSE153855 (dataset 1) and GSE214517 (dataset 2). The newly generated bulk RNA-seq data of INS-1 832/13 cells were deposited to GEO under the accession number GSE263888. The reused human islet scRNAseq dataset was from E-MTAB-5061 (dataset 2). All other data are available in the article and its Supplementary files or from the corresponding author upon request. Source Data are provided with the paper. Source data are provided with this paper.

## Code availability

Code availability: dGCNA is available on https://github.com/Hjerling-Leffler-Lab/dGCNA_Networks.

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

## Acknowledgements

The authors would like to thank the Eukaryotic Single Cell Genomics facility at SciLifeLab, where scRNAseq was performed, the Human Tissue Lab at Lund University Diabetes Centre, and the Nordic Network for Islet Transplantation for providing islets. Rickard Fred for technical assistance, Karen Saylor, and Nancy Martin for animal husbandry and genotyping. The authors are grateful to Claes Wollheim, Patrick F. Sullivan, Patrik Ernfors, and Sten Linnarson for critical reading of the manuscript. This work was funded by joint grants to J.H-L. and N.W. from the Novo Nordisk Foundation (grants NNF15OC0016546 and NNF16OC0021200). In addition, N.W. was supported by the Swedish Research Council (2024-03413, 2020-01017, 2017-00862, and 521-2012-2119), The Regional research foundation (ALF) (2022-Project0191), Novo Nordisk Foundation (NNF24OC0091917, NNF20OC0063916), Diabetes Wellness Research Foundation Sweden, EFSD (EFSD/AZ Cellular Plasticity Programme 2015, EFSD and Lilly European Diabetes Research Programme 2024), The Hjelt Foundation, The Royal Physiographic Society in Lund, The Swedish Diabetes Foundation, and The Påhlsson Foundation. The work at Lund University Diabetes Centre was supported by the Swedish Research Council (Linnaeus grant, Dnr 349-2006-237, and Strategic Research Area Exodiab, Dnr 2009-1039), the Swedish Foundation for Strategic Research (Dnr IRC15-0067). Also, J.H-L. was supported by the Swedish Research Council (awards 2014-3863, 2018-00799, and 2024-03567) and ERC Grant Agreement [819540]. L.E. was supported by the Swedish Diabetes Foundation (DIA2022-723) and the Swedish Research Council (2019-01406). The salary for E.C. was funded by the Swedish Research Council (2020-02179).

## Author contributions

Conceptualization: J.A.M.L., J.H-L., and N.W.; dGCNA algorithm and software, J.A.M.L. and J.H-L.; Formal analysis: J.A.M.L., A.H., A.L., P.C., M.N., S.M.B., P.S., E.C., S.S., D.K., and A.K.; Methodology: J.A.M.L., A.H., A.L., J.H-L., and N.W. Investigation: J.A.M.L., A.H., A.L., A.L.P., L.S., S.C., R.B.P., N.G.S., M.L., A.B.M.M., C.L., P.E., P.C, and N.W. Writing –Original Draft: J.A.M.L., A.L., J.H-L., and N.W. Writing –Review & Editing: All authors. Funding Acquisition: J.H-L., N.W., and R.S. Resources: C.L., P.F.J., L.E., R.S., J.H-L., and N.W.

## Funding

## Competing interests

P.E. is an employee at AstraZeneca. All other authors declare no competing interests.

## Additional information

J.A. Martínez-López[1,2,14], A. Lindqvist[3,14], A. Lopez-Pascual [3,14], A. Harder[1,4], P. Chen [5], M. Ngara[3], L. Shcherbina[3], S. Siffo[3], E. Cowan [6], S. M. Baira[7], D. Kryvokhyzha [6], A. Karagiannopoulos [6], S. Chriett[3], N. G. Skene[1], R. B. Prasad [6], M. Lancien[8], P. F. Johnson [9], P. Eliasson[10], L. Eliasson [6], C. Louvet[8], P. Spégel [7], A. B. Muñoz-Manchado [1,11], R. Sandberg [12], J. Hjerling-Leffler [1,15] ✉ & N. Wierup [3,13,15] ✉

[1]Laboratory of Molecular Neurobiology, Department of Medical Biochemistry and Biophysics, Karolinska Institutet, Stockholm, Sweden. [2]Department of Engineering, Universidad Loyola, Seville, Spain. [3]Lund University Diabetes Centre, Department of Experimental Medical Science, Lund University, Malmö, Sweden. [4]Department of Medical Epidemiology and Biostatistics, Karolinska Institutet, Stockholm, Sweden. [5]Division of Clinical Chemistry, Department of Laboratory Medicine, Karolinska Institutet, Stockholm, Sweden. [6]Lund University Diabetes Centre, Department of Clinical Sciences in Malmö, Lund University, Malmö, Sweden. [7]Department of Chemistry, Centre for Analysis and Synthesis, Lund University, Lund, Sweden. [8]INSERM UMR 1064, Center for Transplantation and Immunology, Université de Nantes, Nantes, France. [9]Mouse Cancer Genetics Program, Center for Cancer Research, NCI, Frederick, MD, USA. [10]Bioscience Cardiovascular, Research and Early Development, Cardiovascular, Renal and Metabolism (CVRM), Biopharmaceuticals R&D, Astra-Zeneca, Gothenburg, Sweden. [11]Departamento de Anatomía Patológica, Biología Celular, Histología, Historia de la Ciencia, Medicina Legal y Forense y Toxicología. Instituto de Investigación e Innovación Biomédica de Cádiz (INiBICA), Universidad de Cádiz, Cádiz, Spain. [12]Department of Cell and Molecular Biology, Karolinska Institutet, Stockholm, Sweden. [13]Scania University Hospital, Clinical Research Centre, Malmö, Sweden. [14]These authors contributed equally: J. A. Martínez-López, A. Lindqvist, A. Lopez-Pascual. [15]These authors jointly supervised this work: J. Hjerling-Leffler, N. Wierup. ✉e-mail: jens.hjerling-leffler@ki.se; nils.wierup@med.lu.se

