## [Transparent Peer Review file · Nature Communications]

Single-cell mRNA-regulation analysis reveals cell type-specific mechanisms of type 2 diabetes

Corresponding Author: Professor Nils Wierup

Version 0:

Reviewer comments:

Reviewer #1

(Remarks to the Author)

The article by Martinez-Lopez describes a network-based analysis tool for deriving meaningful cell-specific and functionally relevant signals from single cell RNA seq data. The dGCNA tool is a valuable knowledge from this article, which is also supported with mechanistic insights into the function of two genes identified with this method.

- 1) How many single cells were from non-T2D and T2D islets within each dataset? Overall, fewer non-T2D cells are seen in figure 1 and also alpha cells are more abundant/selected in sequencing. Can authors shed some light on this as alpha and beta are in similar numbers (around 50%) within islets? Some discussion is beneficial.
- 2) Figure 1: include markers for all 11 groups. Need to write the process of (Conos integration and clustering) for marker selection as shown in Figure 1D.
- 3) Figure 2: it is great to see the workflow, however why the 3 genes in 2A are selected specifically? Instead, it would be informative to show this correlation data for the seven genes that you show later on.
- 4) It is good to generate dGCNA plots with dataset 1 and then check for the overlap. However, considering the small overlap, I wonder if the pathways (Fig 3) would change if you perform dGCNA on dataset 2 and with all data (32 donors) together. This analysis is crucial to gain the confidence in using this tool on different datasets still deriving same pathway/network information. Would you find the genes CEBPG and TMEM176A/B in these two separate analyses?
- 5) Fig 3D: Is DESeq between T2D and non-T2D from dataset 1 or all data? clarify in the legend.
- 6) Although I understand that dGCNA looks at the correlations and altered coefficients between different cells and disease conditions, it would be important to check the expression difference in mRNA and protein levels for the identified genes. Data for six identified genes at protein level is presented. However, is it from non-T2D islets or T2D islets? a comparison for these 7 genes at single cell mRNA and islet protein level between non-T2D and T2D is essential. You need to also provide a comparison for these genes between non-T2D and T2D islets bulk RNA-seq data (publicly available) to confirm the utility of dGCNA in single cell seq for identifying newer regulatory genes.
- 7) Fig S2 is very poor quality.
- 8) What is the rationale to select the 7 genes that are presented and specifically TMEM176A/B? Fig 5 indicates some other genes Gabarap, Spire 1, Atraid perform better than Tmem176a/b. This needs a justification.
- 9) Fig 6I-J: need the cebpg KD (alone, no thapsigargin) data for comparison. Is the data expressed as log fold? Is it log10? Need to specify in legends.
- 10) Fig 6G: where is loading control data?
- 11) Fig 7E: is this comparing WT-chow diet vs double KO-high fat diet? Again, legend does not provide much information. High fat diet composition is not provided as well as how many weeks of feeding? In methods, please include these details along with age, numbers, and sex.
- 12) The choice of these animal models is unclear as these animals are not beta-cell specific KO, whereas the genes identified are from dGCNA analysis of single beta cells. The cell line work is in rat insulinoma/beta cell line, which is logical to mimic the defects seen in T2D single beta cells. Need some discussion on this.
- 13) Figure 8: Is this using all data or dataset 1? It would be equally important to identify genes and pathways that are differently identified in alpha and beta cells using dGCNA.

(Remarks on code availability)

Reviewer #2

(Remarks to the Author)

In the manuscript titled "Single-cell mRNA-regulation analysis reveals cell type-specific mechanisms of type 2 diabetes" the authors developed a differential gene coordinated network analysis (dGCNA) method to analyze single-cell RNA-seq data. It uses gene networks instead of differential gene expression to determine differences between T2D and non-diabetic islets scRNA-seq. A coordinated gene network makes it easy to identify pathways affected by a disease. The motivation and background of the study fill an important gap in the field. However, some issues need to be addressed/clarified as listed below:

Major comments:

1. The authors have divided the human islets into datasets 1 and 2 based on the source. However, part of dataset 2 has already been published. The authors could combine the datasets and validate the pipeline on another independent dataset, previously published or otherwise.
2. Figure 2 is not explained clearly in the result section. It is critical to explain the pipeline clearly. Figure 2A is not clear. The result should explain what the clustering signifies in terms of the analysis. They should explain why these 3 genes were chosen and whether they belong to the same network or not. Figure 2 should be explained in context of biological effects or action.
3. Figure 3A and Figure 8A should be explained in more detail - what does y-axis represent? How are these clusters formed along the gene numbers? Which dataset was used to generate this figure?
4. In Figure 5, the authors shortlisted 7 genes whose functions are not known in beta cells. They should explain whether these genes belong to the same or different networks. It says they are highly ranked, but they are not the highest ranked. What was the basis for choosing these? The explanation should emphasize the significance of the dGCNA method and its applicability in finding new genes in the disease.
5. Figure 5C and D should be combined to compare the effect of IBMX with glucose alone. The basal insulin concentration should also be plotted in the graph in addition to 16.7mM glucose, to see the trend in presence of the siRNAs. The legends should mention the normalization, statistics, and how many times the experiment was repeated.
6. In addition to KIAA1324 and phalloidin, it would be interesting to see the expression of any or a few top-ranked genes in a network in human islets. This would signify the pipeline. Also, microfilament or microtubules have negative network scores. Explanation should be given as to what a positive or negative network score means in terms of gene function/expression/pathway membership etc.
7. In subsequent figures, the authors focused on CEBPG and TMEM176A/B based on their effect on insulin secretion. But other genes, too (GABARAP, SPIRE1, and ATRAID), that showed a higher fold change in insulin secretion. An explanation as to why the genes that were chosen were chosen should be provided so that the readers understand it in context of the utility of the pipeline.
8. Figure 6G, there is no loading control. The protein expression quantification should be normalized by loading control expression in addition to NC.
9. Figure 6I and J CEBPg knockdown alone as a control for the effect of its knockdown on these gene expressions should be included.
10. In the discussion, authors should add limitations of the pipeline, and that beta cell specific KO was not used for CABPg potentially as discussed above.

Minor comments:

1. Provide better resolution images for Figures S1 and S2. The labels in S2 are not readable.
2. For the Seahorse experiment in Figures S4 and S6, all the OCR graphs should be labeled to show which stage it belongs to. The raw Seahorse data and ECAR graph should also be available for interpretation.
3. The authors concluded that Cebpg KD is not primarily exerted at the mitochondria level, but it does alter basal OCR. An explanation should be provided for this result.
4. Add "on high fat diet" in line - "However, Tmedm176a/b knockout mice exhibited reduced beta cell mass (Fig. 7B)...".

(Remarks on code availability)

Reviewer #3

(Remarks to the Author)

The study introduces the differential gene coordination network analysis (dGCNA) as a novel tool for interrogating single-cell RNA sequencing (scRNAseq) data to uncover Type 2 Diabetes (T2D)-associated differential coordination of gene expression in pancreatic islet cells. Importantly, it highlights the power of network-based approaches to uncover functional gene networks and deepen our understanding of T2D pathophysiology.

Strengths:

Innovative Methodology: The introduction of dGCNA represents a significant advancement in the analysis of scRNAseq data. Unlike traditional differential expression analysis, dGCNA enables the identification of coordinated gene expression changes within specific cell types, providing a more robust understanding of T2D pathophysiology. This method proves to be particularly effective in revealing both canonical and non-canonical pathways associated with beta cell dysfunction in T2D.

Cell Type-Specific Insights: The study's focus on cell type-specific changes, particularly in beta and alpha cells, allows for a detailed exploration of how T2D differentially impacts various islet cell types.

Validation Across Datasets: The findings were validated across two independent datasets, strengthening the credibility of the results. This cross-validation, combined with in vitro and in vivo validation in mouse models and human islets, underscores the robustness of the dGCNA method in uncovering biologically relevant gene networks.

Minor comments and areas for improvement:

Limited Alpha Cell exploration: While the manuscript provides some insights into the alterations in alpha cells, the focus remains largely on beta cells. Expanding the investigation into alpha cell dysfunction, particularly given their role in glucagon secretion and glucose regulation, could offer a more comprehensive understanding of the cellular mechanisms in T2D.

Potential for Clinical Translation: While the authors address the exciting potential for therapeutic targets, it would benefit from a more detailed discussion on how these findings could be translated into clinical practice. Specifically, the development of pharmacological agents targeting the newly identified pathways or genes should be explored in future studies.

(Remarks on code availability)

Reviewer #4

(Remarks to the Author)

In this manuscript, Martínez-López et al. developed a computational method for identifying networks of differentially coordinated genes (dGCNA). They applied this method to a single-cell RNA-seq dataset consisting of 8,511 cells from 16 non-diabetic and 16 type 2 diabetic (T2D) islet donors and identify 11 differential gene networks (NDCGs) in beta cells, largely recapitulating known pathways that are dysregulated in T2D in beta cells including unfolded protein response (UPR). They follow up on two genes (CEBPG, TMEM176) and show that they affect insulin secretion in a rat beta cell line through distinct biological mechanisms. They also apply dGCNA to alpha cells, which found 4 NDCGs, albeit with little new biological insight. They conclude that dGCNA is a powerful approach for identifying coordinated networks of genes that affect beta cell function, which include several genes not previously linked to beta cells.

Major comments:

- 1) There are many non-diabetic + type 2 diabetic single-cell datasets that are much larger, both in terms of number of cells and donors (PMID: 38049589, 37582230, 37231096). I would suggest seeing whether including larger datasets enhances your ability to detect NDCGs, especially for cell types such as delta or gamma cells which are understudied.
- 2) How concordant are the genes in NDCGs and those identified through functional screens of insulin content (PMID: 36543916)?
- 3) Heterogeneity in beta cells in T2D pathogenesis has been widely reported (PMID: 37669939, 36928765, 37231096, 34183850). Does dGCNA identify different sets of NDCGs for each functional beta cell state?
- 4) One of the pitfalls of dGCNA is the correlative nature of the analysis and the inability to separate cause from consequence (which is common to any analysis solely based on 'omics data). Are your NDCGs enriched for genes identified from GWAS of T2D, fasting glucose, insulin secretion?
- 5) It has been reported that alpha cells are also functionally impaired in T2D (PMID: 35108513). It would be interesting to expand on the section about alpha cells and have some additional context for your findings.

Minor comments:

- 1) Genes or pathways with known roles in beta cell function or similar phrasing is mentioned multiple times in the text, but there are no references, so it is unclear where these genes are coming from.

(Remarks on code availability)

Version 1:

Reviewer comments:

Reviewer #1

(Remarks to the Author)

Thank you for revising the manuscript taking all comments on board.

(Remarks on code availability)

Reviewer #2

(Remarks to the Author)

All my concerns have been addressed.

(Remarks on code availability)

Reviewer #3

(Remarks to the Author)

(Remarks on code availability)

Reviewer #4

(Remarks to the Author)

The authors have addressed most of my comments in this revision and have added additional analyses for alpha cell modules, and enrichment for relevant GWAS traits and other functional assays. These analyses expand on the value of the method and have added more context to the findings in light of other recent studies reporting similar phenomena in islets.

Minor comment:

I think the limitation around per cell read depth and the power analysis is important to show given that the majority of current single-cell studies are using 10x or Parse kits for sc/snRNA-seq. Please add the power analysis as a supplemental figure.

(Remarks on code availability)

We would like to thank the reviewers for thorough reading and for insightful comments and positive words on our study. We have addressed all points and suggestions for improvements and have revised the manuscript accordingly.

Please see our point-by-point response below. Our comments are in red and changes made in the manuscript are presented in *italics*.

REVIEWER COMMENTS

Reviewer #1 (Remarks to the Author)

The article by Martinez-Lopez describes a network-based analysis tool for deriving meaningful cell-specific and functionally relevant signals from single cell RNA seq data. The dGCNA tool is a valuable knowledge from this article, which is also supported with mechanistic insights into the function of two genes identified with this method.

1) How many single cells were from non-T2D and T2D islets within each dataset? Overall, fewer non-T2D cells are seen in figure 1 and also alpha cells are more abundant/selected in sequencing. Can authors shed some light on this as alpha and beta are in similar numbers (around 50%) within islets? Some discussion is beneficial.

We thank the reviewer for pointing this out. Information on cell number (transcriptomes) is available on p18 in Methods. Dataset 1: 3645 cells. Dataset 2: 4866 cells. For clarity we have now included this information in Table S1, where we have indicated the number of cells for each cell type, data set and disease status.

Regarding the proportion of alpha and beta cells, the reviewer is correct in that we have observed an overrepresentation of alpha cells in our data sets. Interestingly, this has also been reported in other studies using FACS-based scRNAseq protocols (reviewed in PMID 35482056). We have added a comment about this in Discussion on p14.

“It should be mentioned that, in accordance with other FACS-based studies¹³, alpha cells were over-represented in our data sets.”

2) Figure 1: include markers for all 11 groups. Need to write the process of (Conos integration and clustering) for marker selection as shown in Figure 1D.

We thank the reviewer for pointing this out. We have now added markers for all 11 cell populations in Figure 1C. In figure 1D we plot differentially expressed genes in each cluster but for four clusters we recovered too few cells for the algorithm to be able to identify any differentially expressed genes - this is now mentioned in the figure legend:

“No differentially expressed genes were detected for Epsilon cells, Macrophages, Endothelial cells or Mast cells, due to low number of recovered cells”

We have clarified how the integration of data sets 1+2 was done with a separate paragraph in methods page 19. We have also added a paragraph after this specifying how the differential genes in Figure 1C and D were selected.

“Integration of datasets

For datasets 1 and 2, the expression of transcript variants with the same gene name were summed. The datasets were integrated and clustered using Conos and cell types were assigned using label propagation based on previous clustering analysis and known markers. We used the basicP2proc function from Pagoda to preprocess each dataset and facilitate integration. A conos object was created

with `Conos::new()` to analyse the joint dataset. Clustering and dimensional reduction was performed using the `buildGraph` function from `pagoda`. A shared nearest neighbor (SNN) graph was built using principal component analysis (PCA) as the dimensionality reduction method, retaining the top 40 components. Global clusters were detected using the leiden clustering algorithm implemented in the `findCommunities` function from `Pagoda`. The graph was embedded into a two-dimensional space using UMAP for visualization, and markers were selected based on a combination of differential expression and biological expertise.

Identification of cell type specific gene expression

For figure 1C we plotted canonical markers for each cell type in the integrated dataset (1+2). For figure 1D we ran “`getDifferentialGenes`” function in `Conos` and plotted the top 5 Z-scores ordered after specificity of expression. “

3) Figure 2: it is great to see the workflow, however why the 3 genes in 2A are selected specifically? Instead, it would be informative to show this correlation data for the seven genes that you show later on.

This point is well taken, and we have gotten questions about these example genes at oral presentations as well. The purpose of showing these specific genes was to illustrate an example of genes with a striking visual difference in correlation between non-T2D and T2D beta cells. They were thus identified by manual inspection. The correlation for the seven genes is strong when computed but not as easy for the eye to detect. For clarity and to avoid confusion we have thus removed Figure 2A, and replaced the accompanying text from the results section (see below). This also will help with the confusion on which data sets are being analyzed (see below). Furthermore, we have decided to move the workflow part of the figure to Figure S2.

We have replaced the text with the following to introduce the concept of “coordination” of expression (p4):

“Gene regulatory networks are typically computed across cells and cell types to identify genes that are co-expressed. However, coordinated expression between genes within a cell is likely an important aspect of gene regulation and cell biology. The identification of such clusters of genes within a cell type would suggest that single-cell transcriptomes represent snapshots of parallel cellular states captured across different phases. We thus hypothesized that altered coordination on a transcriptome scale represents meaningful information about disease biology in T2D. “

4) It is good to generate dGCNA plots with dataset 1 and then check for the overlap. However, considering the small overlap, I wonder if the pathways (Fig 3) would change if you perform dGCNA on dataset 2 and with all data (32 donors) together. This analysis is crucial to gain the confidence in using this tool on different datasets still deriving same pathway/network information. Would you find the genes `CEBPG` and `TMEM176A/B` in these two separate analyses?

We thank the reviewer for this comment since this is a key aspect of our paper. We apologize that it was unclear in the first version. The reason for this was that the first panel in old figure 2 (old Figure 2A) was only from data set 1. We have removed this part of the figure (see comment above) and the reference to it in the text. The data in Figure 3 (new Figure 2) is indeed from the combined dataset 1 and 2 (32 donors). This combined data set is the basis of most analysis in the paper. The only exception is new Figure 3 (Figure 4 in first version) where we have analyzed the two data sets separately to investigate the robustness by which dGCNA identified perturbed pathways.

We explicitly express the fact that all analysis is based on the merged dataset now on page 4 “*We subsequently analyzed the merged dataset in all analysis except figure 3.*” We have also clarified throughout the manuscript.

To reveal T2D-affected biological pathways in beta cells, we performed dGCNA on the combined data set 1 and 2 (n=32), and the resulting dendrogram indicated a high level of modularity, revealing eleven networks of differentially coordinated genes (NDCGs) pointing to pathway alterations in T2D beta cells (Fig. 2A and Table S2). Each of the NDCGs were associated with highly specific gene ontology (GO) terms (Fig. 2B and Table S3) and were named according to biological functions (either the top GO terms or function identified after literature review)(p4).

We also checked if the seven genes chosen for follow up from the merged data set were present in the two separate analyses: seven were also detected in dataset 1 and five out of seven in dataset 2 (including both *CEPBG* and *TMEM176B*). *DBI* and *TMEM176A* were missing from the modules in dataset 2. The fact that 12/14 observations were reproducible in data sets of halved size speaks to the robustness of the method.

5) Fig 3D: Is DESeq between T2D and non-T2D from dataset 1 or all data? clarify in the legend.

We thank the reviewer for helping us to improve this aspect of clarity of our paper. The data in old Fig 3D (now Fig 2D) is from the combined dataset 1 and 2. This has now been clarified in the legend and text.

6) Although I understand that dGCNA looks at the correlations and altered coefficients between different cells and disease conditions, it would be important to check the expression difference in mRNA and protein levels for the identified genes. Data for six identified genes at protein level is presented. However, is it from non-T2D islets or T2D islets? a comparison for these 7 genes at single cell mRNA and islet protein level between non-T2D and T2D is essential. You need to also provide a comparison for these genes between non-T2D and T2D islets bulk RNA-seq data (publicly available) to confirm the utility of dGCNA in single cell seq for identifying newer regulatory genes.

We thank the reviewer for this suggestion. The IHC images of human islets (now in Figure 5, 6 and Figure S6A) are from non-T2D islets and the purpose was to confirm protein expression in beta cells. We have now added IHC also for *DBI* to Fig S6A.

None of the genes were detected as differentially expressed in our DEG analysis. We have also examined differential expression of the seven genes in the largest bulkRNA datasets published (McDonald humanislets.com PMID: 38948734, islet gene view (<https://mae.crc.med.lu.se/IsletGeneView/>) PMID: 35948367 and Solimena's two cohorts PMID: 29185012). These data are now presented in Table S3. Only two of the genes (*ATRAID* and *SPIRE1*) are differentially expressed in bulk RNAseq data.

We have also added systematic IHC for each of the seven genes in pancreatic sections from both non-T2D and T2D donors of our in-house cohort. Morphometric analyses of beta cell expression is now presented in Figure S6B. Only one of the targets (*DSTN*) show significantly different protein expression in T2D beta cells.

We have added a section on page 7 to describe these analyses:

“To test the predictive power of dGCNA in identifying genes involved in T2D-related processes we selected seven genes (TMEM176A/B (“Microfilaments”), DBI (“Proliferation”), CEPBG (“UPR”), DSTN (“Ribosome”), GABARAP (“Ribosome”), SPIRE1 (“UPR”), ATRAID (“Lysosome”) without a well-known presence or function in human beta cells that were highly ranked in five of the NDCGs (Fig. 2C). For selection criteria see Methods. We confirmed the presence of the protein products in human pancreatic sections from donors in dataset 1 (Fig. 5A, Fig. 6A and Fig S6A) and performed

morphometric analysis of beta cell protein product expression in T2D vs. non-T2D islets revealing increased protein levels of DSTN (Fig. S6B). None of the genes were differentially expressed in our scRNAseq data (Fig. S6C and Table S2) and only SPIRE1 and ATRAID were differentially expressed in two out of four previously published bulk RNAseq datasets^{5,7,46} (Table S4).”

Although these genes were not identified as differentially expressed in bulk RNAseq or our own scRNAseq data we have included functional data supporting important functions in beta cells. This supports the notion that disease-related altered coordination of gene expression is an alternative (and perhaps statistically more robust) path to identifying genes important to T2D.

7) Fig S2 is very poor quality.

We apologize for the low resolution of this figure. This was an effect of file compression. We have now provided a new file with high resolution (new Figure S6).

8) What is the rationale to select the 7 genes that are presented and specifically TMEM176A/B? Fig 5 indicates some other genes Gabarap, Spire 1, Atraid perform better than Tmem176a/b. This needs a justification.

We thank the reviewer for helping us clarify our paper. The purpose of the functional validation experiments was to test whether dGCNA can identify new T2D-affected regulators of beta cell function. Therefore, we focused on genes without previously known expression or function in beta cells.

For target selection we thus had strict criteria:

We selected genes with no previously known expression or function in human beta cells. We systematically performed manual literature review of genes that were close to the tips of the dendrograms. We chose *Tmem176a/B* and *Cebpg* for in vivo studies due to access to mouse knock-out models. We have now added information about the target selection criteria on p21 of Methods.

“Selection of genes for functional validation

We selected genes for functional follow up based on the following criteria: high eigencentality in the RDN from different modules. We then applied filters: 1) no previous publication with search terms “beta cell” or “islet” on PubMed, 2) robust expression in INS-1 832/13 cells (assessed with bulkRNAseq). One exception was made for DBI that had two older papers in hamster cell line and rat islets (7544471 and 1664067). TMEM176A/B and CEBPG were selected for further in vivo studies due to the availability of KO mouse models.”

9) Fig 6I-J: need the cebpg KD (alone, no thapsigargin) data for comparison. Is the data expressed as log fold? Is it log10? Need to specify in legends.

We thank the reviewer for this suggestion. *Cebpg* KD had no effect on UPR genes without the presence of thapsigargin. We have now added these data to Fig S8B and changed the text on p8:

Furthermore, although Cebpg KD had no effect on expression of UPR-mediating genes at basal conditions (Fig S8B), Cebpg KD abolished thapsigargin (UPR-inducer)-induced upregulation of the UPR-mediators Atf4, Atf6 and Xbp1s (Fig. 5I).

The data is presented as log fold change of NC using log10. We have now updated the figures to clearly indicate this and specified this in all legends where this type of analysis was made.

10) Fig 6G: where is loading control data?

We thank the reviewer for finding this. We use stain-free gels and normalize protein expression to total protein content. We apologize for not clarifying this previously. We have now presented an image of the entire membrane illustrating protein loading and normalization factors for each lane in Figure S8A.

We have also clarified this in Methods p27

“Western blotting for Grp78

INS-1 832/13 cells were lysed in RIPA buffer 72h after transfection. Protein content was measured using the BCA Protein assay kit from Pierce. 20 µg of protein was loaded onto Mini PROTEAN TGX stain-free gels (BioRad). Gels were activated using a ChemiDoc MP CCD camera from BioRad. Blotting was performed using a Transblot Turbo Transfer System (BioRad). Membranes were activated and then blocked for 1h using TBS-T with 5% non-fat dry milk. Then, membranes were incubated overnight at 4 °C with Grp78 primary antibodies (#3183S, Cell Signalling), diluted 1:1000 in TBS-T with 5% non-fat dry milk. Membranes were then washed three times in TBS-T, incubated with secondary antibody diluted in TBS-T with non-fat dry milk for 60 min, washed three times in TBS-T, developed using SuperSignal West Pico Maximum Sensitivity Substrate (Thermo Scientific), and imaged using a ChemiDoc MP CCD camera. Proteins were quantified based on total protein normalization with stain-free gels in Image Lab version 6.0.1 (BioRad).“

11) Fig 7E: is this comparing WT-chow diet vs double KO-high fat diet? Again, legend does not provide much information. High fat diet composition is not provided as well as how many weeks of feeding? In methods, please include these details along with age, numbers, and sex.

We thank the reviewer for pointing this out. Data in new Figure 6 D and E is indeed comparing high fat diet fed WT mice and high fat diet fed Tmem176a/b DKO mice. This is now clarified in the legend on p32.

“Reduced beta cell mass (B) and islet size (C) in Tmem176a/b double knockout (DKO) mice (normal diet). Reduced acute insulin response (AIR) (D) and higher glucose levels calculated as iAUC (E) during an IpGTT in Tmem176a/b DKO mice fed a high-fat diet (n=10), compared with WT mice fed a high-fat diet (n=10).”

High fat diet composition is mentioned on p24. Duration of the diet regimen has been added in Methods p24.

(3 weeks of standard rodent chow or high fat diet (60% fat, 20% protein, 20% carbohydrates, Special Diet Services, Witham, UK))

Age, sex, weight and number of animals have been added on p25:

Male Tmem176a/b DKO mice fed HFD (n= 10, age: 15 weeks, weight: 28±2.7g) or a control diet (n=10, age 12 weeks, weight: 25±2.9) and male WT mice fed HFD (n=10, age 14 weeks, weight: 27±0.9) or a control diet (n=10, age 13 weeks, weight: 24.6±2.7) were used.

Number of animals have been added in Legends to Figure 6 and Figure S10.

12) The choice of these animal models is unclear as these animals are not beta-cell specific KO, whereas the genes identified are from dGCNA analysis of single beta cells. The cell line work is in rat

insulinoma/beta cell line, which is logical to mimic the defects seen in T2D single beta cells. Need some discussion on this.

This point is well taken. We of course agree that beta cell-specific KO models would be superior to global KO. We have added the limitation with global KOs to Discussion on p13. The reason for using global KO mouse strains is that they were already available in the labs of collaborators. However, we feel that the data generated in INS-1 cells overall support the findings in the KO models.

P13: "It needs to be mentioned that both the Tmem176a/b DKO and the CEBPG KO mice were global KOs and potential effects in non-islet tissue could be contributing to an insulin-related phenotype. However, our confirmation of a functional role of these genes in beta cells lines suggest that they have important functions in beta cells."

13) Figure 8: Is this using all data or dataset 1? It would be equally important to identify genes and pathways that are differently identified in alpha and beta cells using dGCNA.

Data in Figure 7 (previous Figure 8) is indeed from all 32 donors. This is now clarified in the Results section as well as on page 4 (see answer to comment #4).

P10: "Using the merged dataset, we identified four NDCGs in alpha cells (Fig. 7A)"

We thank the reviewer for the suggestion to expand on the differences between alpha and beta cells. We have added a sentence referencing our Supplementary Figure 11 where we have summarized the findings, and we have expanded the discussion about this on p13-14.

Results: p11:

"A comparison between alpha and beta cells findings are summarized in Figure S11."

Discussion p14:

"We also performed dGCNA on alpha cell data and identified four NDCGs affected in T2D alpha cells (Fig. S9). It should be mentioned that, in accordance with other FACS-based studies¹³, alpha cells were over-represented in our data sets. T2D patients commonly fail to suppress glucagon levels in the postprandial state. This leads to increased hepatic glucose production and aggravated hyperglycemia⁶⁷. As in beta cells, T2D alpha cells displayed a de-coordinated network of glycolysis genes. Our data gain support from previous reports on lower expression of glycolysis genes in alpha cells of T2D donors⁶⁸ and in Gad⁺ non-diabetic donors⁶⁹. Mice lacking the rate limiting enzyme (Gck) of the glycolysis in alpha cells were recently shown to have hampered glucose-induced suppression of glucagon secretion⁵⁹. Furthermore, a hyper-coordinated network with genes related to secretory granules and peptide hormone processing was evident. Disrupted glucose-induced suppression of alpha cell exocytosis of secretory granules has been elegantly demonstrated in T2D alpha cells⁵⁷. In fact, multiple alpha cell dGCNA genes, were overlapping with the alpha cell exocytosis-related genes identified by⁵⁷. Finally, as in beta cells, we identified a de-coordinated network containing mitochondria genes. Mitochondria function is necessary for glucose-induced suppression of alpha cell exocytosis⁵⁷ and hyperglycemia induces alpha cell mitochondrial dysfunction⁷⁰. Thus, our data on perturbed glycolysis, affected secretory granule function, and perturbed mitochondria function are in line with previously observed malfunctioning glucose-induced suppression of glucagon secretion in T2D and provide new insight into the underpinning gene regulation. Contrary to beta cells, in T2D alpha cells a de-coordinated network composed of ribosome subunit genes was evident. This suggests reduced glucagon translation and could be interpreted as a compensatory mechanism, trying to lower glucagon secretion and thereby glucose output from the liver. Furthermore, the alpha cells did in contrast to the beta cells not display any alterations in UPR or cytoskeleton-related gene networks. Thus, our findings confirm

and expand previous knowledge on alpha cell alterations in T2D and highlight cell type-specific alterations in T2D. “

Reviewer #2 (Remarks to the Author):

In the manuscript titled “Single-cell mRNA-regulation analysis reveals cell type-specific mechanisms of type 2 diabetes” the authors developed a differential gene coordinated network analysis (dGCNA) method to analyze single-cell RNA-seq data. It uses gene networks instead of differential gene expression to determine differences between T2D and non-diabetic islets scRNA-seq. A coordinated gene network makes it easy to identify pathways affected by a disease. The motivation and background of the study fill an important gap in the field. However, some issues need to be addressed/clarified as listed below:

Major comments:

1. The authors have divided the human islets into datasets 1 and 2 based on the source. However, part of dataset 2 has already been published. The authors could combine the datasets and validate the pipeline on another independent dataset, previously published or otherwise.

We thank the reviewer for this comment and apologize that our use of the different data set was unclear in the previous version. Please also see our response to reviewer 1 point #4.

The data in all analysis except figure 3 are from the combined datasets 1 and 2. This has now been clarified throughout the manuscript. The corresponding authors started by generating dataset 1. When we were looking for datasets to perform a replication analysis, we contacted author Sandberg who had published parts of dataset 2. He had since obtained additional samples which were included in the new dataset 2. To maximize our power, we decided to use the entire merged data for discovery and the two separate data sets for an analysis of robustness. This is clarified on p3 in Results.

Unfortunately, the dGCNA method is dependent on a large read depth from each individual cell in order to calculate correlation on cellular level. The method works on data sets with smaller read depths for the first steps, but our experience has shown that the topological clustering fails. This unfortunately excludes all other existing larger T2D scRNAseq data sets since they have been performed using single-cell chemistries (like 10xGenomics) that priorities number of cells over reads per cell.

2. Figure 2 is not explained clearly in the result section. It is critical to explain the pipeline clearly. Figure 2A is not clear. The result should explain what the clustering signifies in terms of the analysis. They should explain why these 3 genes were chosen and whether they belong to the same network or not. Figure 2 should be explained in context of biological effects or action.

As mentioned in response to reviewer 1 comment #3 this point is well taken, and we have gotten questions about these example genes at oral presentations as well. The purpose of showing these specific genes was to illustrate an example of genes with a striking visual difference in correlation between non-T2D and T2D beta cells. They were thus identified by manual inspection. For clarity and to avoid confusion we have now removed Figure 2A, and the accompanying text from the results section. This also will help with confusion on which data sets are being analyzed (see below). Furthermore, we have decided to move the workflow panel to Figure S2.

We have replaced the text with the following to introduce the concept of “coordination” of expression on p4:

“Gene regulatory networks are typically computed across cells and cell types to identify genes that are co-expressed. However, coordinated expression between genes within a cell is likely an important aspect of gene regulation and cell biology. The identification of such clusters of genes within a cell type would suggest that single-cell transcriptomes represent snapshots of parallel cellular states captured across different phases. We thus hypothesized that altered coordination on a transcriptome scale represents meaningful information about disease biology in T2D. “

We are unsure what the reviewer means with “context of biological effects or actions”. dGCNA is a computational method that identifies disease related de- and hyper coordination of gene expression using Differential Network Analysis, a type of analysis that has, to our knowledge, previously not been performed on cell type level using single cell data. The results are possibly not as intuitive as differential gene expression, but we hypothesize in our manuscript that hyper-coordination is related to a gain of function and de-coordination is related to loss of function. The novelty of this type of measurement is the reason we have gone to considerable length to biologically validate our findings in the manuscript. We still think that figure 2 (now Figure S2) can be helpful for non-computational readers to understand the mathematics behind the analysis.

3. Figure 3A and Figure 8A should be explained in more detail - what does y- axis represent ? How are these clusters formed along the gene numbers? Which dataset was used to generate this figure?

The y-axis in old Figures 3A and 8A (new Figures 2 and 7) represents the "height" metric from hierarchical clustering, a standard measurement indicating the dissimilarity at which two branches merge. More intuitively, height reflects the minimal level of similarity required for two clusters (branches) to join during hierarchical clustering. A lower height at module definition indicates stricter criteria for genes to cluster, implying stronger co-expression among members of the resulting module. Since dGCNA is based on differential gene-gene correlation, a lower height corresponds to a larger perturbation in gene-gene correlations between the two. Visualizing hierarchical clustering with height on the y-axis and genes (nodes) on the x-axis is standard practice in Weighted Gene Correlation Network Analysis (WGCNA), which is the methodological foundation of our differential gene correlation network analysis). For detailed methodology and interpretation, please see Langfelder and Horvath (2008; <https://pmc.ncbi.nlm.nih.gov/articles/PMC2631488/>) and Langfelder et al. (2008; <https://pubmed.ncbi.nlm.nih.gov/18024473/>).

Regarding the datasets, both figures were generated using the merged data set which has been clarified throughout the manuscript.

4. In Figure 5, the authors shortlisted 7 genes whose functions are not known in beta cells. They should explain whether these genes belong to the same or different networks. It says they are highly ranked, but they are not the highest ranked. What was the basis for choosing these? The explanation should emphasize the significance of the dGCNA method and its applicability in finding new genes in the disease.

-The genes selected were from the following NDCGs: TMEM176A/B (5: Microfilaments), DBI (7: Proliferation), CEBPG (4: UPR), DSTN (1: Ribosome), GABARAP (1: Ribosome), SPIRE1 (4: UPR), ATRAID (3: Lysosome), this is now in addition to Figure 2 also specified in the Results on p7

-We thank the reviewer for helping us clarify our paper. Our aim with the functional validation experiments was to test whether dGCNA can identify new T2D-affected regulators of beta cell function.

We selected genes with no previously known expression or function in human beta cells. We systematically performed manual literature review of genes that were close to the tips of the dendrograms. We have now added information about the target selection criteria on p21 of Methods.

We selected genes for functional follow up based on the following criteria: no previous publication on PubMed with the search terms “beta cell” or “islet”. The only exception was DBI, for which we identified old literature showing an effect of DBI in hamster cell line and rat islets (7544471 and 1664067). Genes were further assessed for potential previously shown relation to diabetes using the search term “diabetes”. DBI and GABARAP showed hits for diabetes, but none with the combined search terms “diabetes AND islet”. In addition, only genes with substantial expression (assessed with bulkRNAseq) in INS-1 832/13 cells were selected. On top of that we selected genes that were close to the tips of the dendrograms. For the first selected target genes (TMEM176A/B, DBI, CEBPG, DSTN, and ATRAID) we did not consider whether the mRNA level was altered in T2D or not. For the following genes (GABARAP and SPIRE1) we actively selected genes that fulfill all the above criteria, but without altered mRNA expression in T2D. TMEM176A/B and CEBPG were also selected due to available KO mouse models.

Our functional validation showed that dGCNA indeed finds new genes in T2D as all selected genes affected insulin secretion or mRNA expression. None of the genes were found to be differentially expressed in our scRNAseq data. Two of them (ATRAID and SPIRE1) were differentially expressed in bulkRNAseq studies (McDonald humanislets.com PMID: 38948734, islet gene view (<https://mae.crc.med.lu.se/IsletGeneView/>) PMID: 35948367 and Solimena’s 2 cohorts PMID: 29185012). These data are now presented in Table S4. Thereby we argue that our data support the usefulness of dGCNA as an alternative to differential gene expression for identifying new genes in disease.

5. Figure 5C and D should be combined to compare the effect of IBMX with glucose alone. The basal insulin concentration should also be plotted in the graph in addition to 16.7mM glucose, to see the trend in presence of the siRNAs. The legends should mention the normalization, statistics, and how many times the experiment was repeated.

We thank the reviewer for this excellent suggestion. We have added the combined figures with 2.8 mM glucose, 16.7 mM glucose and 16.7 mM glucose+IBMX in Figure 3F.

We have also added the requested information to the legends. For statistics we have now added a section in Methods and a Source Data file is now attached with the revised version of the manuscript:

“Statistics

All experimental data were analyzed using 2-way ANOVA, followed by Tukey’s test post hoc, or by student’s t-test. Differences with $p < 0.05$ were considered significant. For all experimental raw data see Source Data file.”

6. In addition to KIAA1324 and phalloidin, it would be interesting to see the expression of any or a few top-ranked genes in a network in human islets. This would signify the pipeline. Also, microfilament or microtubules have negative network scores. Explanation should be given as to what a positive or negative network score means in terms of gene function/expression/pathway membership etc.

We thank the reviewer for this suggestion. We have now performed new immunohistochemistry experiments for the ribosome marker RPS3 in pancreatic specimens T2D and non-T2D donors. Morphometrical analyses confirmed, as predicted by dGCNA, altered ribosome content in T2D beta cells. These data are now presented in Figure 4 E-F.

Related to our top-ranked genes, one of our key messages in the paper is that a gene can be important for T2D even though there is no differential expression between T2D and non-T2D cells. Rather it is the combined organization of gene expression that is altered, likely representing genetic programs for specific cellular tasks (as shown by our systematic GO-term analysis). We have now performed new immunohistochemical experiments and morphometrical analyses for all our target genes. This is presented in Figure S6B.

With regards to “what a positive or negative network score means in terms of gene function/expression/pathway membership etc” we hypothesise that hypercoordination represents a gain of function and de-coordination means loss of function in terms of pathway function. Predicting how such a change (gain or loss) of a genetic program affects individual proteins (actin staining) is beyond the scope of the paper. Thus, the fact that we see more microfilament staining in T2D patients might represent a malfunctioning control of filament organization rather than a gain of function. We have added this to results p12:

“Phalloidin staining of human pancreatic sections confirmed increased beta cell actin density in T2D donors possibly representing a malfunctioning control of filament organization.”

7. In subsequent figures, the authors focused on CEBPG and TMEM176A/B based on their effect on insulin secretion. But other genes, too (GABARAP, SPIRE1, and ATRAID), that showed a higher fold change in insulin secretion. An explanation as to why the genes that were chosen were chosen should be provided so that the readers understand it in context of the utility of the pipeline.

We apologize for that our gene-selection process was unclear. The purpose of the functional validation experiments was to test whether dGCNA can identify new T2D-affected regulators of beta cell function. *TMEM176A/B* and *CEBPG* were thus selected due to availability of KO mouse models. We have added a statement on this to the methods p21:

“TMEM176A/B and CEBPG were also selected due to available KO mouse models.”

8. Figure 6G, there is no loading control. The protein expression quantification should be normalized by loading control expression in addition to NC.

We thank the reviewer for finding this. We use stain-free gels and normalize protein expression to total protein content. We apologize for not clarifying this previously. We have now presented an image of the entire membrane illustrating protein loading and normalization factors for each lane in Figure S8A.

We have also clarified this in Methods p27

“Western blotting for Grp78

INS-1 832/13 cells were lysed in RIPA buffer 72h after transfection. Protein content was measured using the BCA Protein assay kit from Pierce. 20 µg of protein was loaded onto Mini PROTEAN TGX stain-free gels (BioRad). Gels were activated using a ChemiDoc MP CCD camera from BioRad. Blotting was performed using a Transblot Turbo Transfer System (BioRad). Membranes were activated and then blocked for 1h using TBS-T with 5% non-fat dry milk. Then, membranes were incubated overnight at 4 °C with Grp78 primary antibodies (#3183S, Cell Signalling), diluted 1:1000 in TBS-T with 5% non-fat dry milk. Membranes were then washed three times in TBS-T, incubated with secondary antibody diluted in TBS-T with non-fat dry milk for 60 min, washed three times in TBS-T, developed using SuperSignal West Pico Maximum Sensitivity Substrate (Thermo Scientific), and imaged using a ChemiDoc MP CCD camera. Proteins were quantified based on total protein normalization with stain-free gels in Image Lab version 6.0.1 (BioRad).“

9. Figure 6I and J CEBP γ knockdown alone as a control for the effect of its knockdown on these gene expressions should be included.

We thank the reviewer for this suggestion which was also raised by reviewer 1 comment #9. Cebpg KD had no effect on UPR genes without the presence of thapsigargin. We have now added these data to Fig S8B and changed the text on p8:

Furthermore, although Cebpg KD had no effect on expression of UPR-mediating genes at basal conditions (Fig S8B), Cebpg KD abolished thapsigargin (UPR-inducer)-induced upregulation of the UPR-mediators Atf4, Atf6 and Xbp1s (Fig. 5I).

Also Figure 3 D-E shows the effect of Cebpg KD on *Ins1* and *Ins2* without the presence of thapsigargin.

10. In the discussion, authors should add limitations of the pipeline, and that beta cell specific KO was not used for CABP γ potentially as discussed above.

This point is well taken. We of course agree that beta cell specific KO models would be superior to a global. We have added the limitation with global KOs to Discussion on p13. The reason for using global KO mouse strains is that they were already available in the labs of collaborators. However, we feel that the data generated in INS-1 cells overall support the findings in the KO models.

P13: "It needs to be mentioned that both the Tmem176a/b DKO and the CEBPG KO mice were global KOs and potential effects in non-islet tissue could be contributing to an insulin-related phenotype. However, our confirmation of a functional role of these genes in beta cells lines suggest that they have important functions in beta cells."

We have also introduced a limitations section with regards to read depth in the discussion p15:

"An important limitation for the dGCNA pipeline is the read depth required for the full analysis. The two datasets presented consisted of approx. 1M reads/cell. Typical read depth (15k-150k reads/cell) from methods that prioritize number of cells rather than reads/cell does not provide enough power for a successful topological analysis (data not shown)."

Minor comments:

1. Provide better resolution images for Figures S1 and S2. The labels in S2 are not readable.

We apologize for the low resolution of these figures and we have provided new files (Figure S1 and Figure S6).

2. For the Seahorse experiment in Figures S4 and S6, all the OCR graphs should be labeled to show which stage it belongs to. The raw Seahorse data and ECAR graph should also be available for interpretation.

Thank you for finding this. We have now labelled the figures as indicated in legend for new Figure S8 and Figure S10. We have also added the ECAR graphs in new Figure S8 and Figure S10.

All raw data is now presented in the Source Data file attached to the submission.

3. The authors concluded that Cebpg KD is not primarily exerted at the mitochondria level, but it does alter basal OCR. An explanation should be provided for this result.

We thank the reviewer for pointing this out. The observed moderate reduction in basal respiration (Fig S8) is not in line with the observed increase in insulin secretion seen after *Cebpg* KD (Fig 3F). Therefore, we conclude that the effect on insulin secretion is not related to the effect on the mitochondria. We have added a comment on this in results p8:

“Seahorse experiments (Fig. S8E) showed that Cebpg KD, in agreement with elevated ER stress, caused a moderate reduction in basal respiration, acute response, and proton leak (Fig. S8 F, G, J). These observations are not in line with the observed enhanced insulin secretion after Cebpg KD (Fig. 3F and Fig. 5L). But, our results using several insulin secretagogues (including the mitochondria fuel a-KIC) suggest that the effect of Cebpg KD is not primarily exerted at the mitochondria level.”

4. Add “on high fat diet” in line - “However, *Tmedm176a/b* knockout mice exhibited reduced beta cell mass (Fig. 7B)...”.

We apologize if this was not clearly described. In fact, the data in the new Figure 6B is not from mice fed a high fat diet.

This is now clarified in the figure legend: ” **A** *TMEM176B* protein is expressed in human beta cells. **B** Reduced beta cell mass and islet size (**C**) in *Tmem176a/b* double knockout (DKO) mice (normal diet). ”

We do not have beta cell mass data for mice on a high fat diet.

Reviewer #3 (Remarks to the Author):

The study introduces the differential gene coordination network analysis (dGCNA) as a novel tool for interrogating single-cell RNA sequencing (scRNAseq) data to uncover Type 2 Diabetes (T2D)-associated differential coordination of gene expression in pancreatic islet cells. Importantly, it highlights the power of network-based approaches to uncover functional gene networks and deepen our understanding of T2D pathophysiology.

Strengths:

Innovative Methodology: The introduction of dGCNA represents a significant advancement in the analysis of scRNAseq data. Unlike traditional differential expression analysis, dGCNA enables the identification of coordinated gene expression changes within specific cell types, providing a more robust understanding of T2D pathophysiology. This method proves to be particularly effective in revealing both canonical and non-canonical pathways associated with beta cell dysfunction in T2D.

Cell Type-Specific Insights: The study's focus on cell type-specific changes, particularly in beta and alpha cells, allows for a detailed exploration of how T2D differentially impacts various islet cell types.

Validation Across Datasets: The findings were validated across two independent datasets, strengthening the credibility of the results. This cross-validation, combined with in vitro and in vivo validation in mouse models and human islets, underscores the robustness of the dGCNA method in uncovering biologically relevant gene networks.

We thank the reviewer for the positive response to our manuscript.

Minor comments and areas for improvement:

Limited Alpha Cell exploration: While the manuscript provides some insights into the alterations in alpha cells, the focus remains largely on beta cells. Expanding the investigation into alpha cell dysfunction, particularly given their role in glucagon secretion and glucose regulation, could offer a more comprehensive understanding of the cellular mechanisms in T2D.

We thank the reviewer for this suggestion. We have accordingly added enrichment tests for alpha cell modules on page 11 in the results:

“A comparison between alpha and beta cells findings are summarized in Fig. S11.” Interestingly, 119/249 dGCNA genes were previously found to correlate with alpha cell exocytosis in a Patch-Seq study⁶¹. The highest enrichment was seen in the “Ribosome” NDCG with gene sets “Negative (1mM) T2D” ($p= 1.56e-14$) and “Negative (10mM) Healthy” ($p= 1.60e-24$) but also in “Mitochondria” and “Secretory Granules” across gene sets (Fig. S4B and Table S3). Furthermore, “Glycolysis” showed enrichment for GWAS genes associated with fasting glucose ($p\text{-adj}=0.022$) (Fig. S3C and Table S3). As in beta cells, dGCNA results in alpha cells were overlapping between dataset 1 and dataset 2 for genes (Fig. 7D) and GO-terms (Fig. 7E). DEseq2 identified only 16 differentially expressed genes in alpha cells (Table S6).”

We have also broadened the discussion on our alpha cell data and put our findings in a context on p14.

“We also performed dGCNA on alpha cell data and identified four NDCGs affected in T2D alpha cells (Fig. S9). It should be mentioned that, in accordance with other FACS-based studies¹³, alpha cells were over-represented in our data sets. T2D patients commonly fail to suppress glucagon levels in the postprandial state. This leads to increased hepatic glucose production and aggravated hyperglycemia⁶⁷. As in beta cells, T2D alpha cells displayed a de-coordinated network of glycolysis genes. Our data gain support from previous reports on lower expression of glycolysis genes in alpha cells of T2D donors⁶⁸ and in Gad⁺ non-diabetic donors⁶⁹. Mice lacking the rate limiting enzyme (Gck) of the glycolysis in alpha cells were recently shown to have hampered glucose-induced suppression of glucagon secretion⁵⁹. Furthermore, a hyper-coordinated network with genes related to secretory granules and peptide hormone processing was evident. Disrupted glucose-induced suppression of alpha cell exocytosis of secretory granules has been elegantly demonstrated in T2D alpha cells⁵⁷. In fact, multiple alpha cell dGCNA genes, were overlapping with the alpha cell exocytosis-related genes identified by⁵⁷. Finally, as in beta cells, we identified a de-coordinated network containing mitochondria genes. Mitochondria function is necessary for glucose-induced suppression of alpha cell exocytosis⁵⁷ and hyperglycemia induces alpha cell mitochondrial dysfunction⁷⁰. Thus, our data on perturbed glycolysis, affected secretory granule function, and perturbed mitochondria function are in line with previously observed malfunctioning glucose-induced suppression of glucagon secretion in T2D and provide new insight into the underpinning gene regulation. Contrary to beta cells, in T2D alpha cells a de-coordinated network composed of ribosome subunit genes was evident. This suggests reduced glucagon translation and could be interpreted as a compensatory mechanism, trying to lower glucagon secretion and thereby glucose output from the liver. Furthermore, the alpha cells did in contrast to the beta cells not display any alterations in UPR or cytoskeleton-related gene networks. Thus, our findings confirm and expand previous knowledge on alpha cell alterations in T2D and highlight cell type-specific alterations in T2D.”

Potential for Clinical Translation: While the authors address the exciting potential for therapeutic targets, it would benefit from a more detailed discussion on how these findings could be translated into clinical practice. Specifically, the development of pharmacological agents targeting the newly identified pathways or genes should be explored in future studies.

We thank the reviewer for this comment and have added a sentence with this sentiment to the discussion on page 15:

“The increased power of our analysis could potentially have great impact on the future design of perturbation experiments as well as the development of pharmacological agents targeting the newly identified pathways or genes in future studies.”

Reviewer #4 (Remarks to the Author)

In this manuscript, Martínez-López et al. developed a computational method for identifying networks of differentially coordinated genes (dGCNA). They applied this method to a single-cell RNA-seq dataset consisting of 8,511 cells from 16 non-diabetic and 16 type 2 diabetic (T2D) islet donors and identify 11 differential gene networks (NDCGs) in beta cells, largely recapitulating known pathways that are dysregulated in T2D in beta cells including unfolded protein response (UPR). They follow up on two genes (CEBPG, TMEM176) and show that they affect insulin secretion in a rat beta cell line through distinct biological mechanisms. They also apply dGCNA to alpha cells, which found 4 NDCGs, albeit with little new biological insight. They conclude that dGCNA is a powerful approach for identifying coordinated networks of genes that affect beta cell function, which include several genes not previously linked to beta cells.

Major comments:

1) There are many non-diabetic + type 2 diabetic single-cell datasets that are much larger, both in terms of number of cells and donors (PMID: 38049589, 37582230, 37231096). I would suggest seeing whether including larger datasets enhances your ability to detect NDCGs, especially for cell types such as delta or gamma cells which are understudied.

We thank the reviewers for pointing us in the direction of the studies mentioned. We have updated the manuscript to properly cite the work. The first study, Walker *et al.*, Nature 2023 (38049589), has single-cell analysis comparing a treatment with shRNA and not cases vs. controls (this is done in bulk analysis). The second study (Elgamal *et al.*, Diabetes 2023) used the data deposited in HPAP from the Kaestner lab (17 control and 29 T2D), and the third study generated data from 6 controls and 8 T2d (in addition to 8 pre-T2D). While these datasets are of good quality and have already been used to reveal important biology, the chemistry chosen (different versions of 10xGenomics pipelines) provides less sequencing depth (saturates earlier) per cell and is typically performed with 15-20k reads/cell (manufacturer's recommendation) although reasonable yield can be achieved up until around 100k reads/cell. In our study (using 16 controls and 16 T2D), since we applied SmartSeq2, we could sequence each cell deeper (~1M reads/cell). In fact, in our quality control we removed any individual cell with less than 100k reads. This depth of sampling drastically increases the power of the dGCNA – Thus, we unfortunately cannot use the already existing data sets to confirm the method. In fact, this is why we added dataset 2 which was generated independently. This limitation has been added to discussion p15:

“An important limitation for the dGCNA pipeline is the read depth required for the full analysis. The two datasets presented consisted of approx. 1M reads/cell which allowed for the analysis. Typical read depth (15k-150k reads/cell) from methods that prioritize number of cells rather than information per cell does not provide enough power for a successful topological analysis (data not shown).”

2) How concordant are the genes in NDCGs and those identified through functional screens of insulin content (PMID: 36543916)?

We thank the reviewer for this insightful suggestion. We have compared the genes identified with dGCNA with those identified as insulin-content regulators by Anna Gloyn and team and while some genes were overlapping no significant enrichment was observed with any of the modules.

Out of all beta cell dGCNA genes (671 genes), 36 genes were found also in the Gloyn CRISPR screen. We have now included these observations in Results p6:

“36 (of 671) dGCNA genes, including NKX2-2, were found overlapping but there was no significant enrichment for any module.”

3) Heterogeneity in beta cells in T2D pathogenesis has been widely reported (PMID: 37669939, 36928765, 37231096, 34183850). Does dGCNA identify different sets of NDCGs for each functional beta cell state?

This is an interesting suggestion, but unfortunately, we are underpowered to do such an analysis. We hope to return to this in the future when larger data sets are available.

4) One of the pitfalls of dGCNA is the correlative nature of the analysis and the inability to separate cause from consequence (which is common to any analysis solely based on 'omics data). Are your NDCGs enriched for genes identified from GWAS of T2D, fasting glucose, insulin secretion?

We thank the reviewer for this excellent suggestion. We have performed enrichment analysis using the so far largest collections of genes related to T2D risk (PMID: 38374256), fasting glucose (from the T2D AMP portal: <https://t2d.hugeamp.org/>) and insulin secretion (PMID: 39420167). We chose to also include Mahajan et al., Nat Genet, 2022 (35551307) since they included lists of high confidence gene sets. The results are now included on page 6:

“We calculated enrichment of the modules with genes implicated in beta cell function and T2D via GWAS (Tables S3). For T2D risk GWAS⁴¹ we saw an enrichment in the module “Insulin Secretion” ($p\text{-adj}=4.5\times 10^{-6}$) and remarkably when limiting the analysis to 14 high-confidence genes (fine-mapped and missense)⁴² five of the 32 genes in the module still overlapped with those 14 genes (Odds ratio 129; $p\text{-adj}=6.08e^{-8}$) (Fig. S3A-B). For fasting glucose GWAS (<https://t2d.hugeamp.org/>) we observed enrichment for “Glucose response” ($p\text{-adj}=0.018$) and “Insulin secretion” ($p\text{-adj}=9.2e^{-4}$) (Fig. S3C). For insulin secretion GWAS⁴³ there was an enrichment in “Insulin secretion” ($p\text{-adj}=9.6e^{-4}$) (Fig. S3D). Next, we compared the genes identified in beta cells using dGCNA with genes identified as regulators of insulin content in a recent whole genome CRISPR screen in EndoC- β H1 cells⁴⁴. 36 (of 671) dGCNA genes, including NKX2-2, were found overlapping but there was no significant enrichment for any module (Table S3). Furthermore, 103 of the 671 dGCNA genes were previously shown to correlate with beta cell exocytosis in a patch-seq study, and the module “Mitochondria” was significantly ($p\text{-adj}=0.002$) enriched for genes that negatively correlate with exocytosis in T2D⁴⁵ (Fig. S4A and Table S3).”

5) It has been reported that alpha cells are also functionally impaired in T2D (PMID: 35108513). It would be interesting to expand on the section about alpha cells and have some additional context for your findings.

We thank the reviewer for this suggestion. We have accordingly added enrichment tests for alpha cell modules on page 11 in the results:

“A comparison between alpha and beta cells findings are summarized in Fig. S11.” Interestingly, 119/249 dGCNA genes were previously found to correlate with alpha cell exocytosis in a Patch-Seq study⁶¹. The highest enrichment was seen in the “Ribosome” NDCG with gene sets “Negative (1mM) T2D” ($p=1.56e^{-14}$) and “Negative (10mM) Healthy” ($p=1.60e^{-24}$) but also in “Mitochondria” and “Secretory Granules” across gene sets (Fig. S4B and Table S3). Furthermore, “Glycolysis” showed enrichment for GWAS genes associated with fasting glucose ($p\text{-adj}=0.022$) (Fig. S3C and Table S3). As in beta cells, dGCNA results in alpha cells were overlapping between dataset 1 and dataset 2 for genes (Fig. 7D) and GO-terms (Fig. 7E). DEseq2 identified only 16 differentially expressed genes in alpha cells (Table S6).”

We have also broadened the discussion on our alpha cell data and put our findings in a context on p14.

“We also performed dGCNA on alpha cell data and identified four NDCGs affected in T2D alpha cells (Fig. S9). It should be mentioned that, in accordance with other FACS-based studies¹³, alpha cells were over-represented in our data sets. T2D patients commonly fail to suppress glucagon levels in the postprandial state. This leads to increased hepatic glucose production and aggravated hyperglycemia⁶⁷. As in beta cells, T2D alpha cells displayed a de-coordinated network of glycolysis genes. Our data gain support from previous reports on lower expression of glycolysis genes in alpha cells of T2D donors⁶⁸ and in Gad⁺ non-diabetic donors⁶⁹. Mice lacking the rate limiting enzyme (Gck) of the glycolysis in alpha cells were recently shown to have hampered glucose-induced suppression of glucagon secretion⁵⁹. Furthermore, a hyper-coordinated network with genes related to secretory granules and peptide hormone processing was evident. Disrupted glucose-induced suppression of alpha cell exocytosis of secretory granules has been elegantly demonstrated in T2D alpha cells⁵⁷. In fact, multiple alpha cell dGCNA genes, were overlapping with the alpha cell exocytosis-related genes identified by⁵⁷. Finally, as in beta cells, we identified a de-coordinated network containing mitochondria genes. Mitochondria function is necessary for glucose-induced suppression of alpha cell exocytosis⁵⁷ and hyperglycemia induces alpha cell mitochondrial dysfunction⁷⁰. Thus, our data on perturbed glycolysis, affected secretory granule function, and perturbed mitochondria function are in line with previously observed malfunctioning glucose-induced suppression of glucagon secretion in T2D and provide new insight into the underpinning gene regulation. Contrary to beta cells, in T2D alpha cells a de-coordinated network composed of ribosome subunit genes was evident. This suggests reduced glucagon translation and could be interpreted as a compensatory mechanism, trying to lower glucagon secretion and thereby glucose output from the liver. Furthermore, the alpha cells did in contrast to the beta cells not display any alterations in UPR or cytoskeleton-related gene networks. Thus, our findings confirm and expand previous knowledge on alpha cell alterations in T2D and highlight cell type-specific alterations in T2D.”

Minor comments:

1) Genes or pathways with known roles in beta cell function or similar phrasing is mentioned multiple times in the text, but there are no references, so it is unclear where these genes are coming from.

We thank the reviewer for this advice and have revised the manuscript accordingly. Some genes that are very obviously important in different processes, e.g. glycolysis genes remain without references as we feel that from the text it is clear that they were identified using GO-term analysis.

Reviewer #4

Minor comment:

I think the limitation around per cell read depth and the power analysis is important to show given that the majority of current single-cell studies are using 10x or Parse kits for sc/snRNA-seq. Please add the power analysis as a supplemental figure.

We thank the reviewer for raising this point. dGCNA is a descriptive method, describing the structure of the differences between datasets to reveal biology. Unfortunately, a power analysis is not feasible as dGCNA does not have a hard statistical readout/endpoint. The outcome of dGCNA is networks of genes. As we describe in the paper, attempts to do this in more sparse datasets, i.e. 10X, results in a failure of clustering of genes.

We have now rephrased this in Discussion on p 15 for clarity.

It now reads: "Typical read depth (15k-150k reads/cell) from methods that prioritize number of cells rather than information per cell does not provide enough signal for a successful topological analysis."

Reviewer #1

I only wonder if they should add some limitation/discussion,

1) as they noted "Notably, the expression level (mRNA or protein) did not need to be altered for a gene to be identified as disease regulated." what could be the mechanisms for the observed changes for the two selected genes, if the mRNA or proteins are not altered?

This point is well taken and central to the manuscript. dGCNA is a tool that analyses a different aspect of gene regulation. Differential expression is analysed by averaging over many cells of a cell type - what we are suggesting is that the context in which a gene is expressed in individual cells (within the same cell type) matters. We provide arguments for this in terms of computational analysis as well as experimental validation of the predictions from this analysis.

We hypothesize in the discussion on p15-16 that this likely reflects transcriptional timing:

"Although we have not measured a temporal aspect of the gene coordination, we hypothesize that it represents changes in synchronization of gene expression. Interestingly, such a theory would entail that separate biological pathways are controlled with temporal specificity in cells in general."

If this is true - it would suggest that cells attend to different biological processes at different times and that the coordination of programs is important for disease.

We think that the possibly temporal information that dGCNA provides, beyond a list of disease-related genes, is a way to identify the biological pathways in which a gene is active. Today we lack molecular tools for reading out and perturbing gene expression with this temporal resolution which is why we had to rely on knock-down of these genes.

2) I can understand their strategy to select the two specific genes; however, they should add a limitation that "it is possible that their workflow has identified other genes that they could not validate in vivo."

We thank the reviewer for this comment but would respectfully choose to not add such a comment. The reason is that the possibility that not every implicated gene would be possible to functionally confirm is inherent in any study. We have been completely transparent in why and how the genes we confirmed both in vivo and in vitro were chosen.

3) A reason to perform in vivo studies with male mice only.

HFD studies and IPGTT in Tmem176a/b DKOs were performed in males as it is previously described that males are more sensitive to HFD as reviewed in e.g. PMID 29143855. The purpose of the HFD regimen was to stress the beta cells by inducing perturbed glucose tolerance.

We have added a sentence to the methods about this on P24:

"; as previous studies show that males develop more pronounced insulin resistance and beta cell stress compared with females⁷⁹."

*Please ensure that a Source Data file is included with your resubmission. Within the Source Data file, the relevant raw data from each figure or table (in the main manuscript and in the Supplementary Information) should be represented by a single sheet in an Excel document, or a single .txt file or other file type in a zipped folder. Uncropped blots and gel images should be pasted in and labelled with the relevant panel and identifying information such as the antibody used. An example of the Source Data file is available demonstrating the correct format: <https://www.nature.com/documents/ncomms-example-source-data.xlsx>

There was an error in the previous submission, and we apologize for not noting that the Source Data file was missing. This is now uploaded.

The file should be labelled 'Source Data', with the title and a brief description included in your response here, and should be mentioned in all relevant figure legends using the template text below: "Source data are provided as a Source Data file."

We have uploaded source data in a file labelled "Source Data".

We have mentioned "Source data are provided as a Source Data file." in all relevant figure legends.

*In your data availability section, please differentiate between reused datasets and newly generated datasets and provide hyperlinks for each dataset.

We have clarified this and changed the description to:

"The human islet scRNAseq datasets were deposited at GEO and EBI on the following accession numbers GSE153855 (dataset1, unpublished), GSM6607641 (dataset 2, unpublished) and E-MTAB-5061 (dataset 2, published)."